# Clustered gamma-protocadherins regulate cortical interneuron programmed cell death

Walter R Mancia Leon[1†], Julien Spatazza[1†‡], Benjamin Rakela[2], Ankita Chatterjee[1], Viraj Pande[1], Tom Maniatis[3], Andrea R Hasenstaub[4,5], Michael P Stryker[2,5], Arturo Alvarez-Buylla[1,5]*

[1]Department of Neurological Surgery and The Eli and Edythe Broad Center of Regeneration Medicine and Stem Cell Research, University of California, San Francisco, San Francisco, United States; [2]Department of Physiology and Center for Integrative Neuroscience, University of California, San Francisco, San Francisco, United States; [3]Department of Biochemistry and Molecular Biophysics, Columbia University, New York, United States; [4]Department of Otolaryngology-Head and Neck Surgery, University of California, San Francisco, San Francisco, United States; [5]Kavli Institute for Fundamental Neuroscience, University of California, San Francisco, San Francisco, United States

**Abstract** Cortical function critically depends on inhibitory/excitatory balance. Cortical inhibitory interneurons (cINs) are born in the ventral forebrain and migrate into cortex, where their numbers are adjusted by programmed cell death. Here, we show that loss of clustered gamma protocadherins (*Pcdhg*), but not of genes in the alpha or beta clusters, increased dramatically cIN BAX-dependent cell death in mice. Surprisingly, electrophysiological and morphological properties of *Pcdhg*-deficient and wild-type cINs during the period of cIN cell death were indistinguishable. Co-transplantation of wild-type with *Pcdhg*-deficient interneuron precursors further reduced mutant cIN survival, but the proportion of mutant and wild-type cells undergoing cell death was not affected by their density. Transplantation also allowed us to test for the contribution of *Pcdhg* isoforms to the regulation of cIN cell death. We conclude that *Pcdhg*, specifically *Pcdhgc3*, *Pcdhgc4*, and *Pcdhgc5*, play a critical role in regulating cIN survival during the endogenous period of programmed cIN death.

*For correspondence:
alvarezbuyllaa@ucsf.edu

†These authors contributed equally to this work

Present address: ‡ BrainEver, Paris, France

## Introduction

GABAergic cortical inhibitory interneurons (cINs) regulate neuronal circuits in the neocortex. The ratio of inhibitory interneurons to excitatory neurons is crucial for establishing and maintaining proper brain function (*Rossignol, 2011*; *Chao et al., 2010*; *Marín, 2012*; *Hattori et al., 2017*; *Huang et al., 2007*; *Rubenstein and Merzenich, 2003*). Alterations in the number of cINs have been linked to epilepsy (*Dudek and Shao, 2003*), schizophrenia (*Beasley and Reynolds, 1997*; *Hashimoto et al., 2003*; *Enwright et al., 2016*) and autism (*Gao and Penzes, 2015*; *Cellot and Cherubini, 2014*; *Fatemi et al., 2009*). During mouse embryonic development, the brain produces an excess number of cINs and ~40% of those are subsequently eliminated by apoptosis during early postnatal life, between postnatal day (P) 1 and 15 (*Southwell et al., 2012*; *Denaxa et al., 2018*; *Wong et al., 2018*). What makes the death of these cells intriguing is its timing and location. In normal development, cINs are generated in the medial and caudal ganglionic eminences (MGE; CGE) of the ventral forebrain, far from their final target destination in the cortex. cINs migrate tangentially from their site of origin to reach the neocortex, where they

become synaptically integrated and complete their maturation (*Anderson et al., 1997*; *Wichterle et al., 2001*; *Butt et al., 2005*; *Nery et al., 2003*). The ganglionic eminences are also an important source of interneurons in the developing human brain, where migration and differentiation extend into postnatal life (*Hansen et al., 2013*; *Paredes et al., 2016*; *Ma et al., 2013*). How is the final number of cINs regulated once these cells arrive in the cortex?

Since cINs play a pivotal role in regulating the level of cortical inhibition, the adjustment of their number by programmed cell death is a key feature of their development and essential for proper brain physiology. While recent work suggests that activity-dependent mechanisms regulate cIN survival through their connectivity to excitatory neurons (*Wong et al., 2018*; *Denaxa et al., 2018*; *Duan et al., 2020*; *Priya et al., 2018*) studies indicate that cIN survival is mediated by a population-autonomous (or cell-autonomous) mechanism (*Southwell et al., 2012*). Heterochronically transplanted MGE cIN precursors undergo a wave of apoptosis coinciding with their age, which is asynchronous from endogenous cINs. Whereas it is well established that neuronal survival in the peripheral nervous system (PNS) is regulated through limited access to neurotrophic factors secreted by target cells (*Huang and Reichardt, 2001*; *Aloe and Chaldakov, 2013*; *Oppenheim and Carolanne, 2013*), cIN survival is independent of TrkB, the main neurotrophin receptor expressed by neurons of the CNS (*Southwell et al., 2012*; *Rauskolb et al., 2010*). Moreover, the proportion of cINs undergoing apoptosis remains constant across graft sizes that vary 200-fold (*Southwell et al., 2012*). Taken together, this work suggests that cIN developmental death is intrinsically determined and that cell-autonomous mechanisms within the maturing cIN population contribute to the regulation of their survival.

The clustered protocadherins (Pcdh) (*Wu and Maniatis, 1999*) are a set of cell surface homophilic-binding proteins implicated in neuronal survival and self-avoidance in the spinal cord, retina, cerebellum, hippocampus, and olfactory bulb glomeruli (*Ing-Esteves et al., 2018*; *Wang et al., 2002b*; *Lefebvre et al., 2012*, *Lefebvre et al., 2008*; *Katori et al., 2017*; *Mountoufaris et al., 2017*; *Chen et al., 2017*). In the mouse, the Pcdh locus encodes a total of 58 isoforms that are arranged in three gene clusters, alpha, beta and gamma: *Pcdha*, *Pcdhb*, and *Pcdhg* (*Wu et al., 2001*). The *Pcdha* and *Pcdhg* isoforms are each composed of a set of variable exons, which are spliced to three common constant cluster-specific exons (*Tasic et al., 2002*; *Wang et al., 2002a*). Each variable exon codes for the extracellular, transmembrane and most-proximal intracellular domain of a protocadherin protein. The *Pcdhb* isoforms are encoded by single exon genes encoding both extracellular, transmembrane and cytoplasmic domains (*Wu and Maniatis, 1999*). Of the 58 *Pcdh* genes, it has been suggested that a combinatorial, yet stochastic, set of isoforms is expressed in each neuron (*Esumi et al., 2005*; *Kaneko et al., 2006*; *Mountoufaris et al., 2017*), suggesting a source for neuronal diversity in the CNS (*Canzio et al., 2019*). Interestingly, *Pcdhg* genes, and specifically isoforms *Pcdhgc3*, *Pcdhgc4*, and *Pcdhgc5*, are required for the postnatal survival of mice (*Wang et al., 2002b*; *Hasegawa et al., 2016*; *Chen et al., 2012*). Whether Pcdh genes are required for the regulation of cIN elimination remains unknown.

In the present study, we used a series of genetic deletions of the Pcdh gene locus to probe the role of clustered Pcdhs in the regulation of cIN cell death in mice. We show that *Pcdhg*, but not *Pcdha* or *Pcdhb*, are required for the survival of approximately 50% of cINs through a BAX-dependent mechanism. Using co-transplantation of *Pcdhg*-deficient and wild-type (WT) cINs of the same age, we show that they compete for survival in a mechanism that involves *Pcdhg*. Taking advantage of the transplantation assay, we show that removal of the three *Pcdhg* isoforms, *Pcdhgc3*, *Pcdhgc4*, and *Pcdhgc5*, is sufficient to increase cell death of MGE-derived cINs. Three-dimensional reconstructions and patch-clamp recordings indicate that the *Pcdhg* mutant cells have similar morphology, excitability and receive similar numbers of inhibitory and excitatory synaptic inputs compared to wild type cINs. We conclude that cIN cell death is regulated by all or some of the C-isoforms in the *Pcdhg* cluster and that this process is independent of the structural complexity or intrinsic physiological properties of the cell or the strength of its excitatory and inhibitory synaptic inputs.

## Results

### *Pcdhg* expression in developing cINs

Expression of clustered protocadherins (Pcdh) in the brain starts in the embryo and continues postnatally (*Hirano et al., 2012*; *Frank et al., 2005*; *Wang et al., 2002b*; *Kohmura et al., 1998*). RT-

PCR analysis revealed the expression of each of the 58 isoforms in the Pcdh gene locus in the adult cortex (P30) (*Figure 1A*). Of the 58 Pcdh genes, those in the *Pcdhg* cluster are essential for postnatal survival (*Hasegawa et al., 2016*; *Chen et al., 2012*), and are implicated in cell death in the retina and spinal cord (*Lefebvre et al., 2008*; *Prasad et al., 2008*). We, therefore, determined whether *Pcdhg* genes are expressed in cINs during the period of cIN cell death. Using *Gad1*-GFP mice to label GABAergic cINs (*Tamamaki et al., 2003*), we FACS-sorted GFP-positive (GFP+) and GFP-

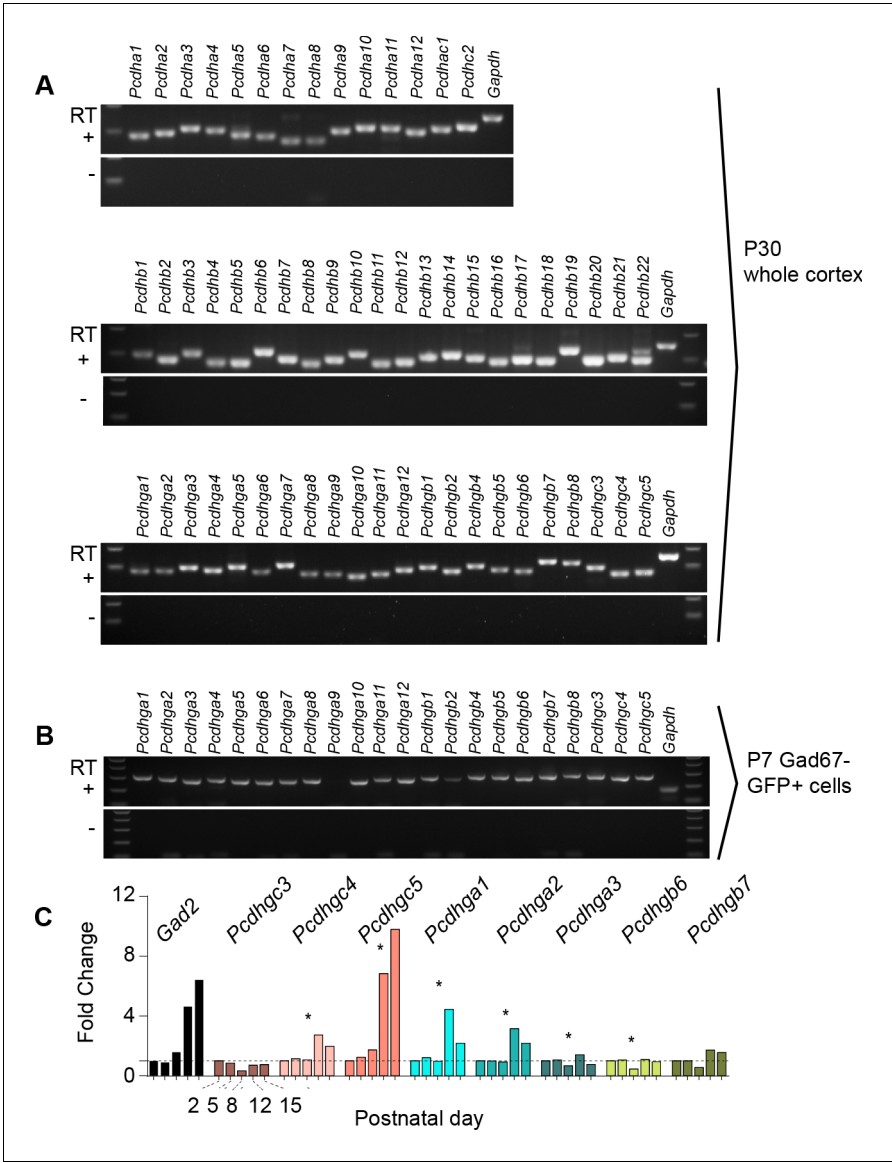

**Figure 1.** Expression of clustered Pcdhs in the mouse cortex and purified cortical GABAergic cells. ( **A**) PCR analysis of clustered *Pcdh* and *Gapdh* gene expression in P30 whole cortex extracts. (**B**) PCR analysis of *Pcdhg* and *Gapdh* gene expression in purified P7 cortical GABAergic cells. (**C**) Quantification of target gene mRNA levels at various postnatal stages (P2, P5, P8, P12, P15) in purified cortical GABAergic cells. P2 mRNA levels used as a reference for each gene (Kruskal-Wallis test, P value = 0.0007 [ *Pcdhgc4*], P value < 0.0001 [*Pcdhgc5*], P value = 0.015 [*Pcdhga1*], P value = 0.024[*Pcdhga2*], P value = 0.003[*Pcdhg*a3], P value = 0.038[*Pcdhgb6*]; n = 3 technical replicas]. Significant p values are marked with*. See *Figure 1—source data 1* for followup of comparisons.

The online version of this article includes the following source data and figure supplement(s) for figure 1:

**Source data 1.** *Pcdhg* expression in cortex.

**Figure supplement 1.** GABAergic markers are enriched in GFP positive FACS-sorted cells from *Gad1*-GFP mice.

negative (GFP-) cells from P7 mice at the peak of cIN cell death (*Figure 1—figure supplement 1A*). We confirmed that GABAergic cell markers (*Gad1, Gad2*) were enriched in the GFP+ population, while markers of excitatory neurons (*Tbr1, Satb2, Otx1*), astrocytes (*Gfap, Aldh1l1*), and oligodendrocytes (*Olig2, Mbp*) were enriched in the GFP- population (*Figure 1—figure supplement 1B*). With the exception of *Pcdhga9* isoform, we detected the expression of all other 21 *Pcdhg* in cINs (*Figure 1B*). To determine the expression pattern of *Pcdhg* at different stages during the period of cell death, we measured the expression level of 8 *Pcdhg* mRNAs (*Pcdhgc3, Pcdhgc4, Pcdhgc5, Pcdhga1, Pcdhga2, Pcdhga3, Pcdhgb6,* and *Pcdhgb7*) at P2, P5, P8, P12 and P15 using qPCR (*Figure 1C*). All eight isoforms were expressed in cINs at each of the five ages studied. Interestingly, the expression of *Pcdhgc5* increased dramatically between P8 and P15. An increase in expression of *Pcdhg* isoforms *Pcdhga1, Pcdhga2* and *PcdhgC4* was also observed at P12, compared to other ages, but this increase was less pronounced than that observed for *Pcdhgc5*. The above results show that all Pcdh isoforms are expressed in cINs and that the expression of *Pcdhg* isoforms *Pcdhga1, Pcdhga2* and *PcdhgC4* and *Pcdhgc5* increases during the period of postnatal cell death.

## Reduced number of cINs in the cortex of *Pcdhg* mutants

Most cINs are produced between embryonic days (E) 10.5 and 16.5 by progenitors located in the medial and caudal ganglionic eminences (MGE and CGE) (*Anderson et al., 1997*; *Wichterle et al., 2001*; *Nery et al., 2002*; *Miyoshi et al., 2010*). To address the potential role of *Pcdhg* in cIN development, we used the *Pcdhg* conditional allele (*Pcdhg^{fcon3}*) to block production of all 22 *Pcdhg* isoforms (*Lefebvre et al., 2008*). In the *Pcdhg^{fcon3}* allele, the third common exon shared by all *Pcdhg* isoforms contains the sequence coding for GFP and is flanked by loxP sites (*Lefebvre et al., 2008*; *Figure 2A*). In unrecombined *Pcdhg^{fcon3}* mice, all *Pcdhg* isoforms are thus fused to GFP. However, when these animals are crossed to a Cre driver line, expression of the entire *Pcdhg* cluster is abolished in Cre-expressing cells (*Prasad et al., 2008*). Robust GFP expression was detected throughout the brain in E13.5 embryos, including cells in the MGE and CGE (*Figure 3B*), indicating expression of *Pcdhg* isoforms in cIN progenitors. We crossed *Pcdhg^{fcon3}* mice to *Gad2^{Cre}* mice (*Taniguchi et al., 2011*) to conditionally ablate all *Pcdhg* in GABAergic cells throughout the CNS at an early embryonic stage (E10.5) (*Katarova et al., 2000*). Recombined cells were visualized thanks to the conditional Ai14 (tdTomato) reporter expression (*Figure 2A*). Heterozygous *Gad2^{Cre}*;Ai14; *Pcdhg^{fcon3/+}* mice were viable and fertile. However, homozygous *Gad2^{Cre}*;Ai14 ;*Pcdhg^{fcon3/fcon3}* mice displayed growth retardation after birth, a hind limb paw-clasping phenotype when held by the tail and were infertile (*Figure 2B*). Brain size as well as cerebral cortex thickness of homozygous *Gad2^{Cre}*;Ai14;*Pcdhg^{fcon3/fcon3}* was similar to those of control mice (*Figure 2B'*). However, the density of tdTomato positive cells in somatosensory and visual cortex was roughly halved in homozygous *Gad2^{Cre}*;Ai14;*Pcdhg^{fcon3/fcon3}* animals, compared to wild type and heterozygous littermates (*Figure 2C and C'*). The density of cINs stained positive for parvalbumin (PV) and somatostatin (SST) (MGE-derived), vasoactive intestinal peptide (VIP )(CGE-derived) or reelin (RLN) (derived from both the MGE and CGE) was significantly reduced in the visual cortex of homozygous *Gad2^{Cre}*;Ai14; *Pcdhg^{fcon3/fcon3}* mice (*Figure 2D* and *Figure 2—figure supplement 1*). Taken together, these experiments indicate that the embryonic loss of *Pcdhg* function in GABAergic progenitor cells leads to a drastically reduced number of cINs in the neocortex, affecting all cIN subtypes similarly.

The developmental defects observed in *Gad2^{Cre}*;Ai14;*Pcdhg^{fcon3/fcon3}* mutant mice may indirectly affect the survival of cINs in a non-cell autonomous manner. We thus decided to restrict the *Pcdhg* loss of function to MGE/POA (preoptic area) progenitors by means of the *Nkx2.1^{Cre}* mouse (*Xu et al., 2008*). MGE/POA progenitors give rise to the majority of mouse cINs, including PV and SST interneurons. NKX2.1 expression is detected in the ventral telencephalon from E9.5 (*Sandberg et al., 2016*; *Shimamura et al., 1995*) and is downregulated in most cINs as they migrate into the developing neocortex (*Nóbrega-Pereira et al., 2008*). *Pcdhg^{fcon3}* mice were crossed to *Nkx2.1^{Cre}* mice. As described above, tdTomato expression was again used to visualize the recombined cells (*Figure 3A*). Homozygous *Nkx2.1^{Cre}*;Ai14;*Pcdhg^{fcon3/fcon3}* embryos lost GFP expression specifically in in the MGE and the preoptic regions (*Figure 3B*), consistent with full recombination and loss of *Pcdhg* function in cells derived from the Nkx2.1 lineage.

At P30, the number of MGE-derived tdTomato+ cells in *Nkx2.1^{Cre}*;Ai14;*Pcdhg^{fcon3/fcon3}* mice was dramatically reduced (~50%) in both the visual and somatosensory cortex (*Figure 3C and C'*). MGE-derived PV and SST interneuron number was similarly reduced in these animals. However CGE-

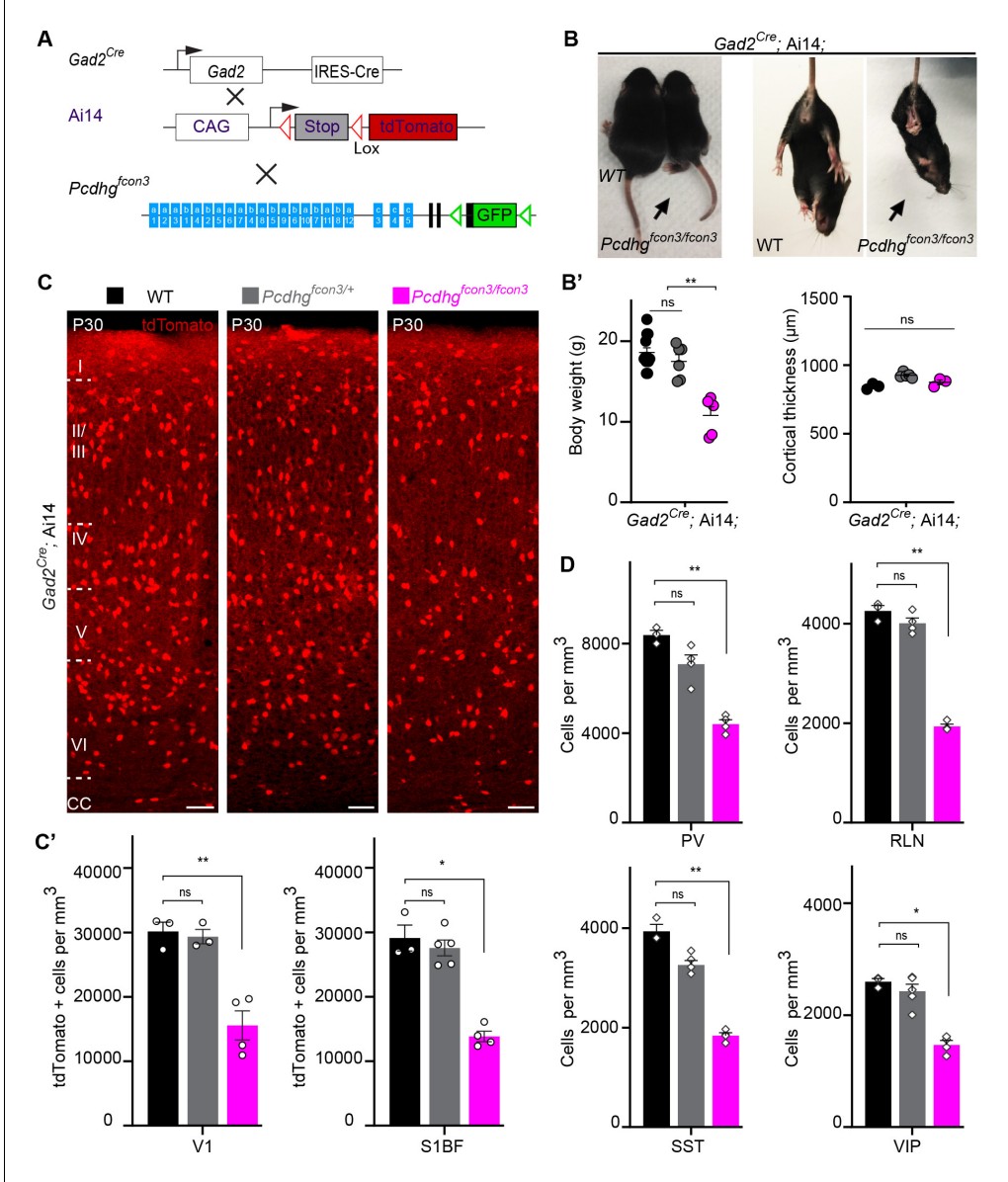

**Figure 2.** Reduced number of GABAergic cINs in *Pcdhg*-deficient mice. (**A**) Mutant mice with loss of *Pcdhg* in GABAergic neurons were generated by crossing conditional *Pcdhg^fcon3* mice to Pan-GABAergic Cre driver (*Gad2*) mice. The conditional Ai14 reporter was used to fluorescently label *Gad2*-expressing cells. (**B**) Photographs of P21 *Gad2^Cre*;Ai14 mice that are wild type (WT) or mutant (*Pcdhg^fcon3/fcon3*) for *Pcdhg*. (**B'**) Body weight and cortical thickness measurements in P30 *Gad2^Cre*;Ai14;*Pcdhg^{+/+}* (*Pcdhg* WT), *Gad2^Cre*;Ai14;*Pcdhg^fcon3/+* (*Pcdhg* HET), and *Gad2^Cre*;Ai14;*Pcdhg^fcon3/fcon3* (*Pcdhg* mutant) mice(Kruskal-Wallis test, P value=0.0027, adjusted p values **p=0.0017, n = 12 mice [*Pcdhg* WT], n = 7 mice [*Pcdhg* HET] and n = 5 mice [*Pcdhg* mutant]). (**C**) Photographs of coronal sections in primary visual cortex (V1) of P30 *Gad2^Cre*;Ai14; *Pcdhg* WT (left), *Pcdhg* HET (middle) and *Pcdhg* mutant (right) mice. All cortical layers are similarly affected (***Figure 2—figure supplement 2***). Scale bar, 100 μm. (**C'**) Quantifications of tdTomato+ cell density in V1 and somatosensory (S1BF) cortex of P30 *Gad2^Cre*;Ai14 *Pcdhg* WT(black), *Pcdhg* HET (grey), and *Pcdhg* mutant (magenta) mice (Kruskal-Wallis test; for V1 (P value = 0.006), for S1BF (P value = 0.009); adjusted p values **p=0.0180, *p=0.036, n = 3–5 mice of each genotype). (**D**) Quantifications of cIN subtype density in V1 cortex at P30. All four non-overlapping cIN subtypes (PV, SST, RLN, and VIP) were similarly reduced in numbers in *Gad2^Cre*;Ai14;*Pcdhg^fcon3/fcon3* mice (*Pcdhg* mutant, magenta) compared to WT controls (Kruskal-Wallis test; for PV (P value = 0.0002), for SST (P value=0.0021), for RLN (P value = 0.0012), and for VIP (P value=0.0093); adjusted p values **p=0.004 (PV), **p=0.0073(SST), **p=0.0093(RLN), *p=0.0365 (VIP), n = 3–5 mice of each genotype).

The online version of this article includes the following source data and figure supplement(s) for figure 2:

**Source data 1.** Quantification of GABAergic cINs, body weight and cortical thickness measurements in controls and *Pcdhg* deficient mice.
**Figure supplement 1.** Reduced number of GABAergic cIN subtypes in *Pcdhg*-deficient mice.
**Figure supplement 2.** Reduced number of GABAergic cINs across cortical layers in *Pcdhg*-deficient mice.
**Figure supplement 2—source data 1.** Quantification of all GABAergic cINs by cortical layer in the visual cortex of controls and *Pcdhg-deficient* mice.

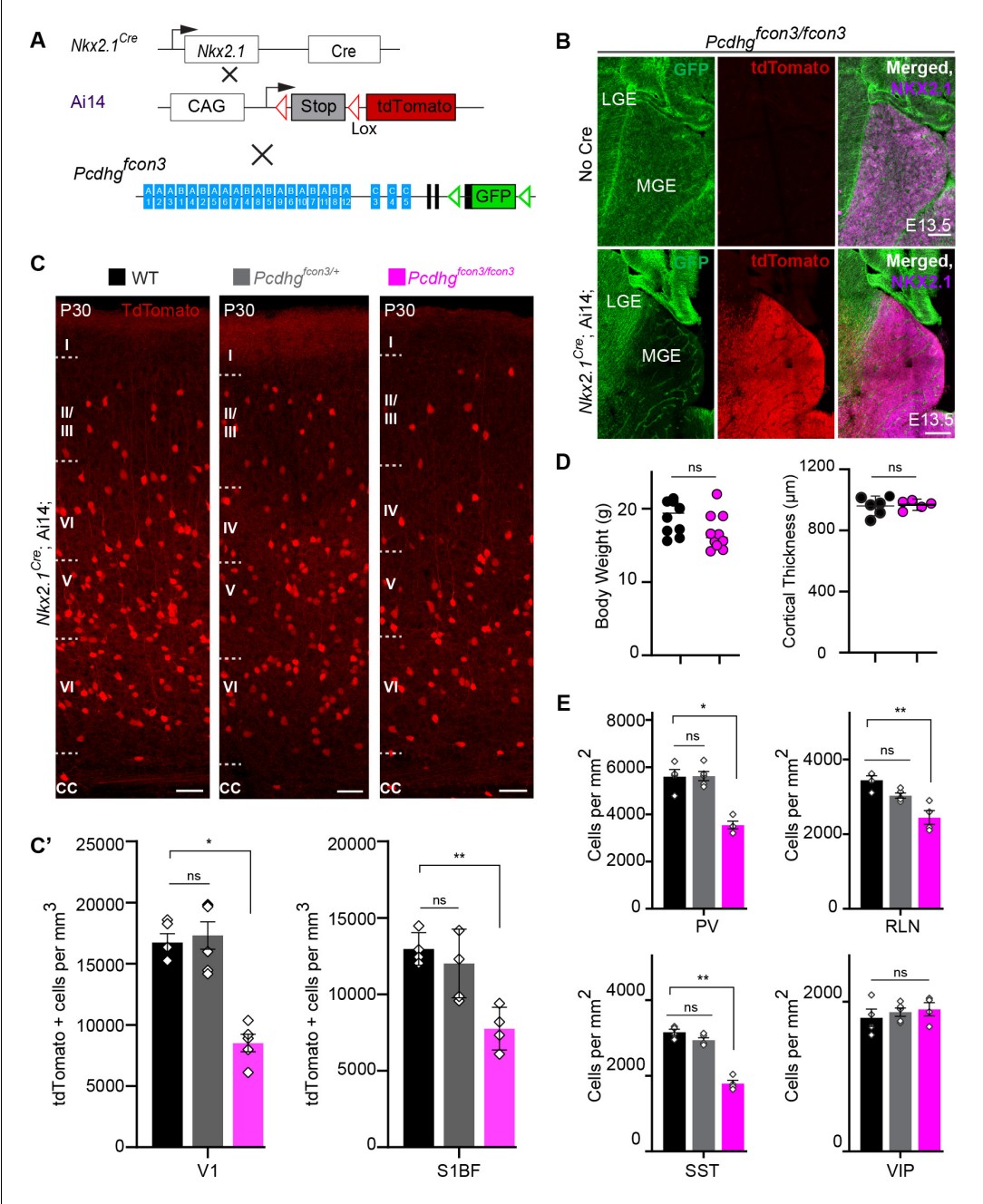

**Figure 3.** Loss of *Pcdhg* genes targeted to Nkx2.1 expressing cells results in selective loss of cIN derived from the MGE. ( **A**) Mutant mice with loss of *Pcdhg* in MGE-derived cIN were generated by crossing *Pcdhg^fcon3* mice to Nkx2.1^Cre mice. The conditional Ai14 line was used to fluorescently label MGE-derived cells. (**B**) *Pcdhg^fcon3/fcon3* mice (at E13.5; top panels) carrying the *Pcdhg* mutant allele, but not Cre, show robust expression of GFP in the MGE. In contrast, in *Nkx2.1^Cre;Ai14;Pcdhg^fcon3/fcon3* mice (at E13.5; bottom panels), carrying the *Pcdhg* mutant allele and expressing Cre, GFP expression was eliminated from the MGE. Note NKX2.1 staining (magenta) in the panels on the right delineates MGE/POA (preoptic area). The few cells left expressing GFP in the MGE are blood vessels and are tdTomato negative. (**C**) Photographs of coronal sections of the primary visual cortex (V1) in *Nkx2.1^Cre;Ai14;Pcdhg^{+/+}* (*Pcdhg* WT), *Nkx2.1^Cre;Ai14;Pcdhg^fcon3/+* (*Pcdhg* HET) and *Nkx2.1^Cre;Ai14;Pcdhg^fcon3/fcon3* (*Pcdhg* mutant). Scale bar, 100 μm. (**C'**) Quantification of the density of tdTomato+ cells in V1 and S1BF cortex of P30 *Nkx2.1^Cre;Ai14;Pcdhg* WT (black), *Pcdhg* HET (grey) and *Pcdhg* mutant (magenta) mice. The number of Nkx2.1-derived cells was significantly reduced in *Nkx2.1^Cre;Ai14;Pcdhg* mutant mice compared to WT controls (Kruskal-Wallis test; for V1(P value=0.002), for S1BF (P value=0.0065), adjusted p values *p=0.0232, **p=0.0168 (S1), n = 4–6 mice of each genotype). (**D**) Body weight and cortex thickness measurements in *Nkx2.1^Cre;Ai14;Pcdhg* WT (black) and *Pcdhg* mutant (magenta) mice at P30. Body weight and cortical thickness were not significantly affected by loss of *Pcdhg* (Mann-Whitney test, body weight (p=0.0547, n = 10 mice of each genotype), cortical thickness (p=0.2857, n = 4–5 mice of each genotype). (**E**) Quantification of tdTomato+ cIN subtypes in V1 mouse cortex at P30. *Nkx2.1^Cre;Ai14;Pcdhg*

*Figure 3 continued on next page*

*Figure 3 continued*

mutant mice (magenta) had significantly reduced numbers of MGE-derived parvalbumin (PV)+, somatostatin (SST)+, and Reelin (RLN)+ cells compared to WT controls. In contrast VIP+ cells, which are derived from the CGE, were not significantly affected (Kruskal-Wallis test; for PV (P value = 0.0113), for SST (P value=0.0009), for RLN (P value = 0.0014), and for VIP (P value=0.636); adjusted p values *p=0.0113 (PV), **p=0.0055 (SST), **p=0.0055 (RLN), n = 4–5 mice of each genotype).

The online version of this article includes the following source data and figure supplement(s) for figure 3:

**Source data 1.** Quantification of Nkx2.1-derived cINs, body weight and cortical thickness measurements in controls and *Pcdhg*-deficient mice.

**Figure supplement 1.** Reduced number of MGE cIN subtypes after loss of *Pcdhg* in Nkx2.1-derived cells.

**Figure supplement 2.** Numbers of Nkx2.1-derived RLN positive cINs are reduced in layers II-VI in *Pcdhg*-deficient mice.

**Figure supplement 2—source data 1.** Quantification of Reelin positive cINs by cortical layer in the visual cortex of controls and *Pcdhg* loss of function mice.

**Figure supplement 3.** Cortical layer distribution of Nkx2.1-derived cINs in *Pcdhg* WT and mutant mice.

**Figure supplement 3—source data 1.** Quantification of Nkx2.1-derived cINs by cortical layer in the visual cortex of controls and *Pcdhg* loss of function mice.

**Figure supplement 4.** Increased survival of non-Nkx2.1-derived SST and PV cINs in *Pcdhg*-deficient mice.

**Figure supplement 4—source data 1.** Quantification of non-Nkx21-derived PV and SST positive cINs in the visual cortex of controls and *Pcdhg* loss of function mice.

derived VIP interneuron density was similar to that of control animals (*Figure 3E* and *Figure 3—figure supplement 1*). A smaller, but significant reduction in the RLN positive cIN population was observed, in agreement with the notion that a subpopulation of RLN cells is born in the MGE (*Miyoshi et al., 2010*). Consistently, layer 1 RLN+ cells, which are largely derived from the CGE (*Miyoshi et al., 2010*), were not affected by *Pcdhg* loss of function, but RNL cells in deeper layers 2–6 (many of which are MGE-derived and also positive for SST) showed reduced numbers (*Figure 3—figure supplement 2*). Interestingly, we observed that in our *Nkx2.1^{Cre}*;Ai14;*Pcdhg^{fcon3/fcon3}* mice, the number of un-recombined PV and SST (PV+/tdTomato- and SST+/tdTomato-) cells was significantly increased compared to WT mice (*Figure 3—figure supplement 4*), a result consistent with recent observations (*Carriere et al., 2020*). PV+/tdTomato- and SST+/tdTomato- cells are likely derived from the most dorsal MGE at the interface with LGE expressing NKX6.2 in a region of low, or no expression of NKX2.1 (*Hu et al., 2017a*; *Hu et al., 2017b*; *Fogarty et al., 2007*; *Sousa et al., 2009*). We do not know if the presence of the conditional *Pcdhg^{fcon3}* allele results in increased production of these cells or if un-recombined cells from this domain increase their survival in compensation for the loss of cINs that lack *Pcdhg* function. If the latter is true, the behavior of these un-recombined PV and SST cINs differs from that observed for WT cells co-transplanted with MGE cells lacking *Pcdhg* function (see below). Together the above results show that embryonic loss of *Pcdhg* function in Nkx2.1-positive progenitors results in a significant reduction in the number of MGE/POA-derived cINs.

## *Pcdhg* function is not required for the proliferation and migration of cIN precursors

The reduction in the number of cINs in *Nkx2.1^{Cre}*;Ai14;*Pcdhg^{fcon3/fcon3}* mice was not a result of abnormal cortical thickness or abnormal layer distribution, as these measures were similar across genotypes in P30 mice (*Figure 3C*). Next, we asked whether migration or proliferation defects in the cIN progenitor population could lead to a reduced cIN density in *Pcdhg* mutant mice. Quantification of the number of dividing cells in the ventricular or subventricular zones at E13.5 and E15.5, using the mitotic marker Phosphohistone H3 (PH3), showed no difference in the number of mitotic cells in the MGE between *Nkx2.1^{Cre}*;Ai14; *Pcdhg^{fcon3/fcon3}* mice and controls (*Figure 4A and B*). Migration of young cINs into cortex was also not affected in *Nkx2.1^{Cre}*;Ai14;*Pcdhg^{fcon3/fcon3}* mice . The tdTomato+ cells in the cortex displayed a similar migratory morphology in *Nkx2.1^{Cre}*;Ai14; *Pcdhg^{fcon3/fcon3}* embryos and controls. Consistent with the absence of an effect of *Pcdhg* on cIN migration, the number of migrating cells in cortex in the marginal zone (MZ), the subplate (SP), and the intermediate and subventricular zone (IZ/SVZ) was equivalent between *Pcdhg* mutant embryos and controls at E15.5 (*Figure 4C and D*). These findings indicate that loss of *Pcdhg* did not affect the proliferation of MGE progenitors or the migration of young MGE-derived cINs into the developing neocortex.

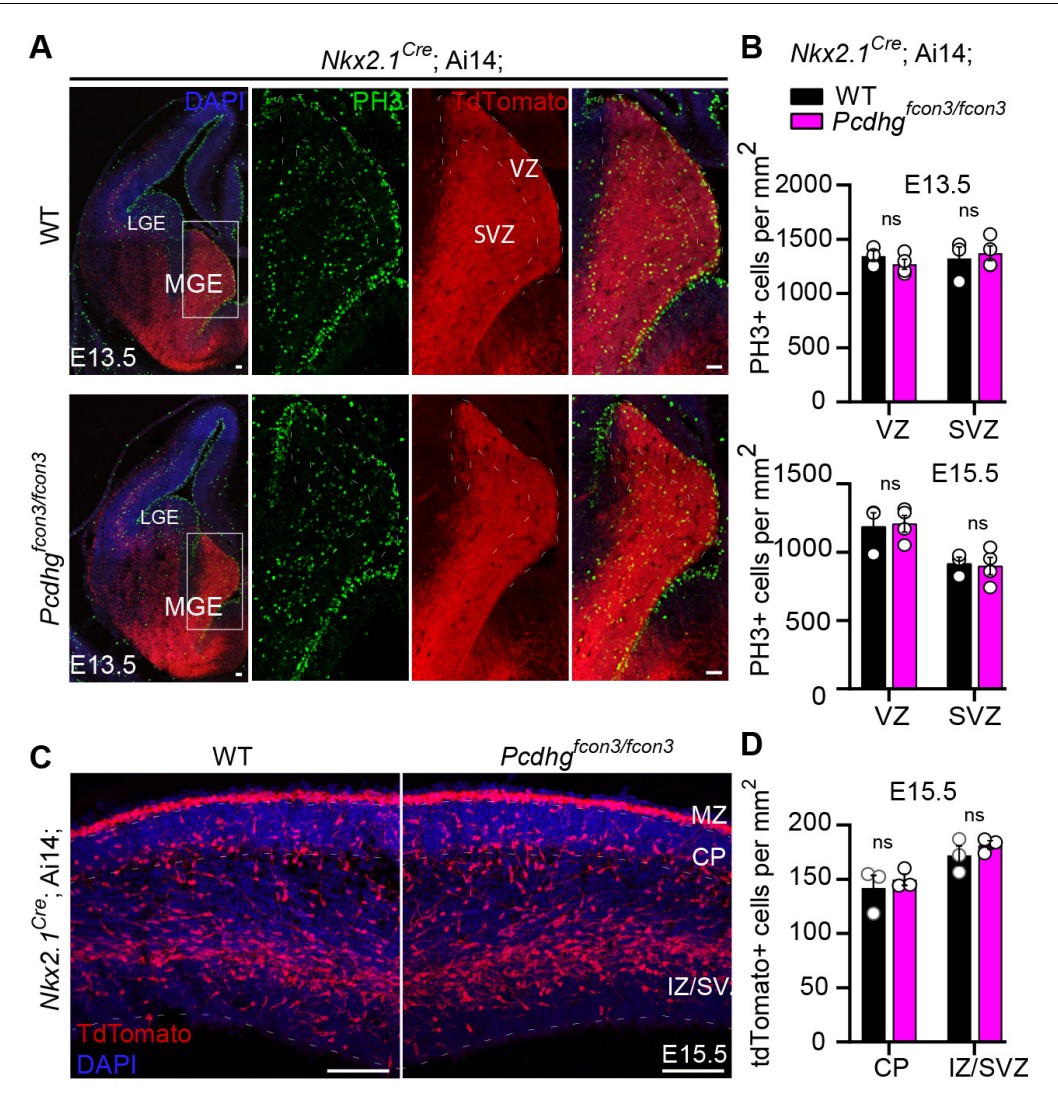

**Figure 4.** Proliferation and migration are not affected by the loss of *Pcdhg* in NKX2.1 expressing cells. (**A**) Photographs of coronal sections through the embryonic forebrains of E13.5 *Nkx2.1^Cre*;Ai14;*Pcdhg^+/+* (*Pcdhg* WT, top panels) and *Nkx2.1^Cre*;Ai14;*Pcdhg^fcon3/fcon3* (*Pcdhg* mutant, bottom panels). Close-up photographs of the MGE from *Nkx2.1^Cre*;Ai14;*Pcdhg* WT (right panels) and *Pcdhg* mutant (bottom, right panels) embryos. Robust reporter activity (tdTomato) was observed in the MGE. Dividing cells were labeled using the mitotic marker PH3. Note that the size and number of PH3+ cells in the MGE was similar in the mutant and control brains. Scale bars, 50 μm. (**B**) Quantification of PH3+ cells from MGE ventricular (VZ) and subventricular zone (SVZ) in E13.5 (top) and E15.5 (bottom) *Nkx2.1^Cre*;Ai14;*Pcdhg* WT (black bars) and *Pcdhg* mutant (magenta bars) embryos (Mann-Whitney test, p=0.4000 (E13.5 VZ), p=0.8571 (E13.5 SVZ), p=0.8571 (E15.5 VZ), p>0.999 (E15.5 SVZ), n = 3–4 embryos of each genotype). (**C**) Photographs of coronal sections of dorsal cortex at E15.5 showing the migrating MGE-derived cIN in *Nkx2.1^Cre*;Ai14;*Pcdhg* WT (left) and *Pcdhg* mutant (right) embryos. Note the robust migratory streams of young neurons in the SVZ and in the marginal zone (MZ). From these regions, cells disperse into the intermediate zone (IZ) and cortical plate (CP). Similar numbers of migrating cIN were observed in mutants and controls. Scale bar, 100 μm. (**D**) Quantifications of number of migrating MGE-derived cINs in the CP and in the IZ/SVZ of *Nkx2.1^Cre*; Ai14;*Pcdhg* WT (black) and *Pcdhg* mutant (magenta) mice. No significant differences were detected in the number of tdTomato+ migrating cells in *Pcdhg* mutant and WT controls (Mann-Whitney test, p>0.990 (E15.5 CP), p=0.7000 (E15.5 IZ-SVA), n = 3 embryos of each genotype).

The online version of this article includes the following source data for figure 4:

**Source data 1.** Quantification of PH3 positive cells in the embryonic MGE and number of Nkx2.1-derived cINs in the embryonic dorsal cortex of controls and *Pcdhg* loss of function mice.

## Accentuated cIN cell death in *Pcdhg* mutants

A wave of programmed cell death eliminates ~40% of the young cINs shortly after their arrival in the cortex (*Southwell et al., 2012*; *Wong et al., 2018*). This wave starts at ~P0, peaks at P7, and ends at ~P15. Next, we asked if the reduced cIN density observed in *Pcdhg* mutant mice could stem from a heightened number of mutant cINs undergoing apoptosis at the normal time. Such cells were immunolabeled using an antibody directed against cleaved-Caspase 3 (CC3). Since CC3 positive cells are relatively rare, our analysis was performed throughout the entire neocortex, at P0, 3, 7, 10, and 15. Similarly to their wild type littermates, *Nkx2.1$^{Cre}$*;Ai14;*Pcdhg$^{fcon3/fcon3}$* homozygous mice displayed a wave of programmed cell death peaking at P7 (*Figure 5A and B*). However, *Pcdhg* mutant mice had significantly higher numbers of tdTomato+/CC3+ cells compared to controls. We also examined the proportion of CC3+ cells that were tdTomato negative (un-recombined cells that would notably include pyramidal cells, CGE-derived cINs, and glial cells). With the exception of a small, but significant increase observed at P0, we found no significant difference in the number of CC3+/tdTomato- cells between genotypes (*Figure 5B*, bottom graph). This suggests that the survival of neighboring *Pcdhg*-expressing cells is not impacted by the loss of *Pcdhg*-deficient MGE/POA-derived cINs. Importantly, the homozygous deletion of the pro-apoptotic Bcl-2-associated X protein (BAX) rescued cIN density in the *Pcdhg* mutant mice to levels similar to those observed in control *Bax$^{-/-}$*;*Pcdh$^{fcon3/+}$* mice or in mice carrying only the *Bax* mutation (*Bax$^{-/-}$*) (*Southwell et al., 2012*; *Figure 5C*). The above results indicate that loss of *Pcdhg* in MGE/POA-derived cIN enhances their demise through programmed cell death during the developmental period when these cells are normally eliminated.

## Loss of *Pcdhg* does not affect survival of cINs after the period of programmed cell death

We then asked whether *Pcdhg*-expression is also required for the survival of cINs past the period of programmed cell death. To address this question we took advantage of the *PV$^{Cre}$* transgene (*Hippenmeyer et al., 2005*) that becomes activated specifically in PV interneurons starting at around ~P16 (*Figure 6* and *Figure 6—figure supplement 1*). Quantifications of tdTomato+ cell density in *PV$^{Cre}$*;Ai14;*Pcdhg$^{fcon3/fcon3}$* and *PV$^{Cre}$*;Ai14 mice at P60-P100 revealed no significant differences between homozygous and control mice (V1 and S1BF) (*Figure 6D and E*). In contrast, the *Sst$^{Cre}$* line, like *Nkx2.1$^{Cre}$*, induces recombination at embryonic stages. The *Sst$^{Cre}$* allele in Ai14; *Pcdhg$^{fcon3/fcon3}$* reduced the density of SST interneurons measured at P30 to the same level as was found using the *Nkx2.1$^{Cre}$* line (*Figure 6A–C*). Together, our results demonstrate that *Pcdhg* loss of function reduces cIN survival specifically during the endogenous period of cIN cell death, resulting in a reduced cortical density of cINs.

## *Pcdha* and *Pcdhb* do not affect cIN survival

Previous studies indicate that Pcdhs form tetrameric units that include members of the alpha, beta, and gamma clusters (*Schreiner and Weiner, 2010*; *Aye et al., 2014*). We, therefore asked whether *Pcdha* and *Pcdhb* genes also contribute to cIN cell death. Mice that carry a conditional deletion of the entire alpha cluster (*Pcdha$^{acon/acon}$*) were crossed to the *Nkx2.1$^{Cre}$*;Ai14 line, resulting in removal of the *Pcdha* genes specifically from MGE/POA progenitor cells (*Figure 7A*). *Nkx2.1$^{Cre}$*;Ai14;*Pcdha$^{acon/acon}$* mice were viable, fertile, and displayed normal weight (*Figure 7B*, top graph). cIN density in the visual cortex of *Nkx2.1$^{Cre}$*;Ai14;*Pcdha$^{acon/acon}$* mice at P30 was similar to that of *Nkx2.1$^{Cre}$*;Ai14 mice (*Figure 7B*). To determine whether *Pcdhb* genes affected MGE/POA-derived cIN survival, constitutive *Pcdhb gene* cluster knockout (*Pcdhb$^{del/del}$*) mice were crossed to *Nkx2.1$^{Cre}$*;Ai14 mice (*Figure 7A*). Mice carrying a deletion of the entire *Pcdhb* cluster were viable, fertile and of normal weight (*Figure 7C*, top graph) (*Chen et al., 2017*). The density of cINs was similar between mice lacking *Pcdhb* and controls (*Figure 7C*). The above results indicate that unlike the *Pcdhg* cluster, which is essential for the regulation of cIN elimination, the function of *Pcdha* or *Pcdhb* is dispensable for the survival of MGE/POA-derived cINs.

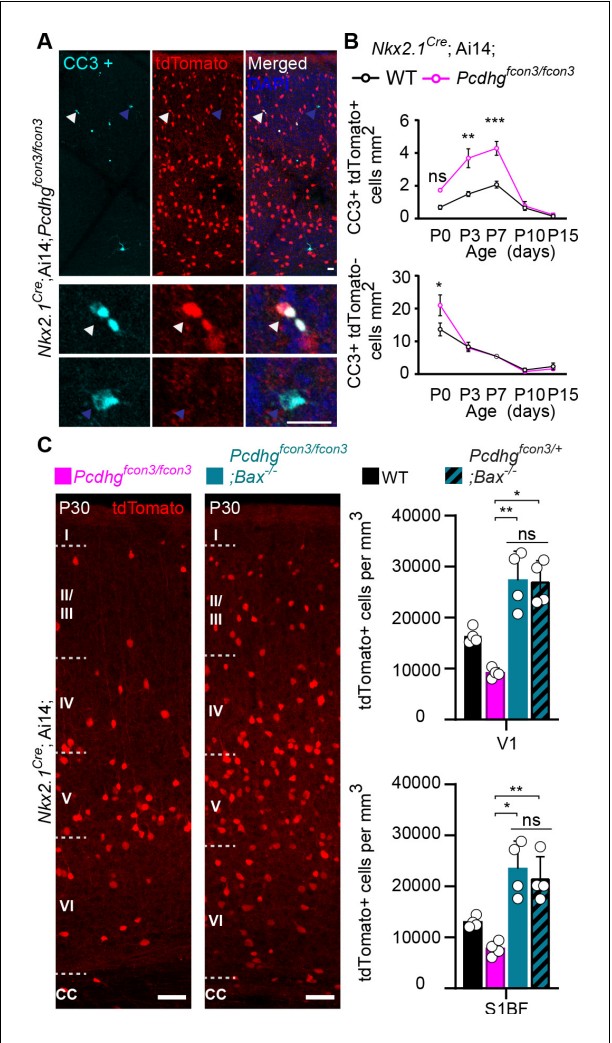

**Figure 5.** Increased programmed cell death in *Pcdhg* mutants is rescued in *Pcdhg-bax* null animals. (**A**) Photographs of coronal sections through a *Nkx2.1^Cre^*;Ai14;*Pcdhg^fcon3/fcon3^* (*Pcdhg* mutant) P7 mouse cortex (top), showing tdTomato+ cINs and cleaved caspase 3 positive cells (CC3+). Close-up photographs (bottom) of tdTomato+, CC3+ (white Arrowheads) and tdTomato-, CC3+ (blue Arrowheads) cells. Scale bar 25 μm. (**B**) Quantification of the density of tdTomato+,CC3+ (MGE-derived, top graph) cells from *Nkx2.1^Cre^*;Ai14;*Pcdhg* WT (black line) and *Pcdhg* mutant (magenta line) mice. Quantification of the density of tdTomato-,CC3+ (non-MGE-derived, bottom graph) cells from *Nkx2.1^Cre^*;Ai14;*Pcdhg* WT (black line) and *Pcdhg* mutant (magenta line) mice. Note that the number of CC3+ cells was significantly increased in the MGE-derived population in *Pcdhg* mutant mice, and coincides with the normal period of programmed cell death for cINs in WT mice (Each age was analyzed with a nested 1-way ANOVA (mouse ID nested within genotype), P value<0.0001. Significant comparisons are marked with *; *p=0.0004 **p=0.0014, ***p=0.0009, n = 3–5 mice of each genotype). (**C**) Coronal sections through the primary visual cortex (V1) of *Nkx2.1^Cre^*;Ai14;*Pcdhg^fcon3/fcon3^* (*Pcdhg* mutant, left) and *Nkx2.1^Cre^*;Ai14; *Pcdhg^fcon3/fcon3^*;*Bax^-/-^* (*Pcdhg* mutant, *Bax* null, right) mice at P30. Quantifications of the density of cINs in V1 (top) and S1BF (bottom) cortex. Note that genetic removal of *Bax* in both *Pcdhg^fcon3/+^* (*Pcdhg* HET) and *Pcdhg^fcon3/fcon3^* (*Pcdhg* mutant) mice rescues cell death to similar levels (Kruskal-Wallis test, P value<0.001 (for V1 an S1BF), adjusted p values for V1 (*p=0.01 , **0.005 ) and for S1BF (**p=0.0109 , *p=0.0286); n = 4–5 mice of each genotype).

The online version of this article includes the following source data for figure 5:

**Source data 1.** Analysis of cIN programmed cell death in controls, *Pcdhg* mutant and *Bax* null mice.

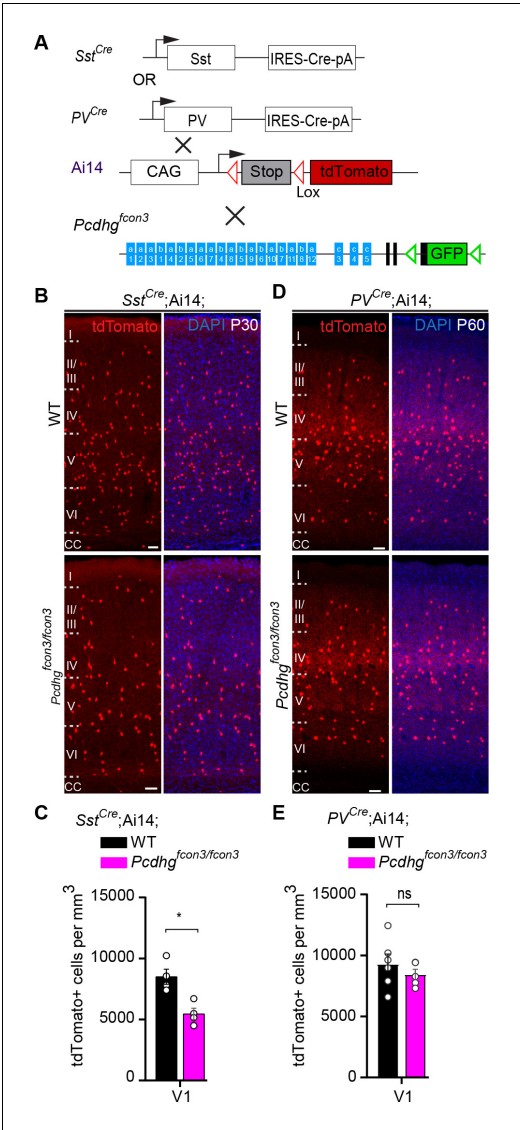

**Figure 6.** *Pcdhg* function is not required for the survival of PV cINs after the period of programmed cell death. (**A**) Mutant mice with loss of *Pcdhg* in SST or PV cells were generated by crossing conditional *Pcdhg*$^{fcon3}$ mice to mice carrying Cre under *Sst* (*Sst*$^{Cre}$) or *Pvalb* (Parvalbumin, *PV*$^{Cre}$). The conditional Ai14 line was used to fluorescently label SST or PV cells. (**B**) Photographs of coronal sections of the primary visual cortex (V1) of P30 *Sst*$^{Cre}$; Ai14; *Pcdhg*$^{+/+}$ (*Pcdhg* WT, top left) and *Sst*$^{Cre}$;Ai14;*Pcdhg*$^{fcon3/fcon3}$ (*Pcdhg* mutant, bottom left) mice. Scale bars, 50 μm. (**C**) Quantifications of the density of tdTomato+ cINs in V1 cortex of *Pcdhg* WT (black) and *Pcdhg* mutant (magenta) *Sst*$^{Cre}$;Ai14 mice at P30 (Mann-Whitney test, **p=0.0286, n = 4 mice of each genotype). (**D**) Photographs of coronal sections of V1 in *PV*$^{Cre}$;Ai14;*Pcdhg*$^{+/+}$ (*Pcdhg* WT, top right) and *PV*$^{Cre}$;Ai14;*Pcdhg*$^{fcon3/fcon3}$ (*Pcdhg* mutant, bottom right) mice at P60. Scale bars, 50 μm. (**E**) Quantifications of the density of tdTomato+ cIN in V1 cortex of *Pcdhg* WT and *Pcdhg* mutant *PV*$^{Cre}$;Ai14 mice at P60-100 (Mann-Whitney test, p=0.4206,n = 5 mice of each genotype).

The online version of this article includes the following source data and figure supplement(s) for figure 6:

**Source data 1.** Analysis of PV and SST- derived cINs at P30 in controls and *Pcdhg* mice.

**Figure supplement 1.** Late postnatal expression of Parvalbumin in cINs.

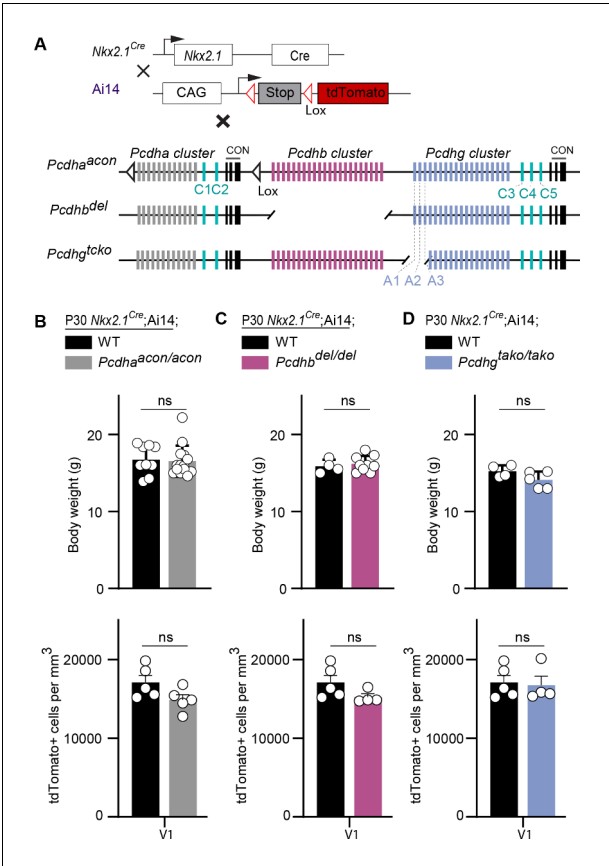

**Figure 7.** Loss of *Pcdha*, *Pcdhb*, or *Pcdhga1, Pcdhga2, and Pcdhga3* genes does not affect the survival of MGE-derived cINs. (**A**) Mutant mice with loss of *Pcdha*, *Pcdhb* or *Pcdhga1, Pcdhga2, and Pcdhga3* genes in MGE-derived cINs were generated by crossing *Pcdha*acon, *Pcdhb*del, and *Pcdhg*tako mice to the *Nkx2.1*Cre mouse line. The conditional Ai14 line was used to fluorescently label MGE-derived cells. (**B**) Measurements of body weight (top graph) in P30 *Nkx2.1*Cre;Ai14;*Pcdha*+/+ (*Pcdha* WT, black bar) and *Nkx2.1*Cre;Ai14;*Pcdha*acon/acon (*Pcdha* mutant, grey bar) mice (Mann-Whitney test, p=0.545, n = 9–14 mice of each genotype). Quantification of the density of MGE-derived cINs (bottom graph) in primary visual cortex (V1) of *Pcdha* WT (black bar) and *Pcdha* mutant (grey bar) P30 mice (Mann-Whitney test, p=0.9603, n = 4–5 mice of each genotype). (**C**) Measurements of body weight (top bar) in P30 *Nkx2.1*Cre;Ai14;*Pcdhb*+/+ (*Pcdhb* WT, black bar) and *Nkx2.1*Cre;Ai14;*Pcdhb*del/del (*Pcdhb* mutant, pink bar) mice (Mann-Whitney test, p=0.712, n = 4–9 mice of each genotype). Quantification of the density of MGE-derived cIN (bottom graph) in primary visual cortex (V1) of *Pcdhb* WT (black bar) and *Pcdhb* mutant (pink bar) P30 mice (Mann-Whitney test, p=0.1111, n = 4–5 mice of each genotype). (**D**) Measurements of body weight (top graph) in *Nkx2.1*Cre;Ai14;*Pcdhg*+/+ (*Pcdhg* WT, black) and *Nkx2.1*Cre;Ai14;*Pcdhg*tako/tako (*Pcdhga1, Pcdhga2, and Pcdhga3* mutant, blue bar) P30 mice (Mann-Whitney test, p=0.175, n = 4–5 mice of each genotype). Quantification of the density of MGE-derived cINs (bottom graph) in in primary visual cortex (V1) of *Pcdhg* WT (black bar) and *Pcdhga1, Pcdhga2, and Pcdhga3* mutant (blue bar) P30 mice (Mann-Whitney test, p=0.9048, n = 4–5 mice of each genotype).

The online version of this article includes the following source data for figure 7:

**Source data 1.** Analysis of Nkx2.1-derived cINs in P30 control, *Pcdha*, *Pcdhb* and *Pcdhg* mutant mice.

## Loss of *Pcdhg* does not affect cIN dispersion after transplantation but affects their survival

In order to compare the timing and extent of migration, survival, and maturation of cINs of different genotypes within the same environment, we co-transplanted into the cortex of host animals, MGE-derived cIN precursor cells expressing red and green fluorescent proteins. MGE cIN precursors were derived from E13.5 *Gad1*-GFP embryos (*Pcdhg* WT controls) or from *Nkx2.1*Cre;Ai14 embryos that were either *Pcdhg* WT or *Pcdhg* mutant (*Figure 8A*). We first confirmed that MGE cells WT for *Pcdhg*, but carrying the two different fluorescent reporters, displayed no differences in their survival.

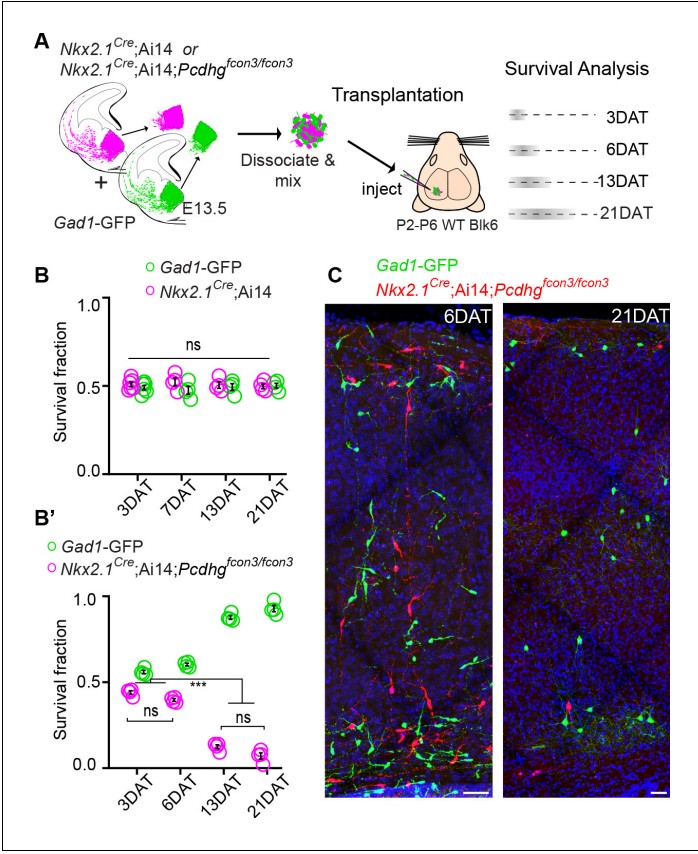

**Figure 8.** *Pcdhg* are required for cIN survival after transplantation. (**A**) Schematic of co-transplantation of MGE-derived cIN precursors. MGE cells were derived from *Nkx2.1^Cre^;Ai14;Pcdhg^+/+^* (*Pcdhg* WT) or *Nkx2.1^Cre^;Ai14; Pcdhg^fcon3/fcon3^* (*Pcdhg* mutant) embryos. These cells were mixed in equal proportions with MGE cells from *Gad1*-GFP embryos (*Pcdhg* WT, green) and transplanted into WT black (Blk) six host recipient mice. Cell survival was analyzed before (3 DAT) and throughout the period of cell death (6–21 DAT). (**B,B'**) Survival fraction of co-transplanted MGE-derived cIN precursors. (**B**) MGE cells were derived from *Gad1*-GFP (green) and *Nkx2.1^Cre^;Ai14;* (magenta) embryos; both GFP+ and tdTomato+ cells carry WT *Pcdhg*. In this control experiment the survival fraction was similar for both genotypes carrying the different fluorescent reporters (2-way ANOVA, F_genotype = 2.54, P value > 0.999; n = 4–6 mice per time point from two transplant cohorts). (**B'**) MGE cells were derived from *Gad1*-GFP WT (green) and *Nkx2.1^Cre^;Ai14;Pcdhg* mutant (magenta) embryos. GFP+ and tdTomato+ cells showed dramatic differences in their survival; the majority of cells carrying the *Pcdhg* mutant allele (magenta) were eliminated between 6 and 21 DAT (2-way ANOVA, F_genotype = 2738.02, P value < 0.0001; adjusted p values ***p<0.0001; n = 4–5 mice per time point from two transplant cohorts. Quantifications in (**B and B'**) were done at 3, 6, 13 and 21 DAT and are represented as fractions of GFP+ or tdTomato+ cells from total cells (GFP + tdTomato+) per brain section. The increase in the proportion of WT cells during this period is not a reflection of increased cell numbers (WT cIN also undergo elimination by programmed cell death (See *Figure 8—figure supplement 1*), but rather that WT cells account for a larger fraction of all transplant-derived cells (WT + *Pcdhg* mutant). (**C**) Representative photographs of cortical sections from transplanted host mice at 6 (left) and 21 (right) DAT. Transplanted MGE cells were derived from *Gad1*-GFP (*Pcdhg* WT, green) and *Nkx2.1^Cre^;Ai14;Pcdhg^fcon3/fcon3^* (*Pcdhg* mutant, red) embryos. Scale bars, 50 μm.

The online version of this article includes the following source data and figure supplement(s) for figure 8:

**Source data 1.** Survival of transplanted MGE-derived cIN precursor cells carrying WT or mutant *Pcdhg*.

**Figure supplement 1.** Number of cIN drops for both the *Pcdhg* WT and *Pcdhg* mutant transplanted population.

**Figure supplement 1—source data 1.** Quantification of transplanted MGE-derived cIN precursor cells carrying WT or mutant *Pcdhg*.

Equal proportions of *Gad1*-GFP cells (*Pcdhg* WT GFP+) and *Nkx2.1^Cre^*;Ai14 cells (*Pcdhg* WT tdTomato+) were co-transplanted into the neocortex of neonatal recipients. While equivalent numbers of red and green cells were mixed before being transplanted, the absolute number of cells transplanted varied from transplant to transplant. In order to compare the survival, we use the fraction of green or red cells, among all co-transplanted cells (red + green). The fraction of surviving GFP+ and tdTomato+ cells at 3, 6, 13, and 21 days after transplantation (DAT) was measured (*Figure 8A and B*, top graph). The contribution of each cell population to the overall pool of surviving cells was found to be ~50% at 3 DAT, and remained constant at 6, 13, and 21 DAT (*Figure 8B*, top graph). This experiment indicates that the fluorescent reporters (GFP or tdTomato) or breeding background does not affect the survival of MGE cINs in this assay. Next, we co-transplanted equal numbers of *Gad1*-GFP cells (*Pcdhg* WT) and *Nkx2.1^Cre^*;Ai14;*Pcdhg^fcon3/fcon3^* cells (*Pcdhg* mutant) into the cortex of WT neonatal recipients. As above, we measured the proportion of surviving GFP+ and tdTomato+ cells at 3, 6, 13, and 21 DAT (*Figure 8A*). Similar numbers of GFP+ and tdTomato+ cells were observed at 3 and 6 DAT. However, the fraction of *Pcdhg* mutant cINs (tdTomato+) surviving was dramatically lower when the transplanted cells reached a cellular age equivalent to that of endogenous cINs after the normal wave of programmed cell death (6DAT is roughly equivalent to P0; 21DAT is roughly equivalent to P15) (*Southwell et al., 2012*). Note that in this experiment the proportion of WT cells increases during this same period. This change in proportion is not a reflection of increased survival, as these cells also undergo elimination by programmed cell death (see below) (*Figure 8—figure supplement 1*), but that, with the increased loss of mutant cells, the WT cells account for a larger fraction of the total.

We next determined whether the survival of *Pcdhg* WT (GFP+) or *Pcdhg* mutant (tdTomato+) cINs was affected by their density (*Figure 9*). At 6 DAT, WT and *Pcdhg* mutant MGE-derived cells had migrated away from the injection site establishing a bell-shaped distribution of density as a function of tangential distance from the injection site (*Figure 9B and B'*). The dispersion of developing cINs lacking *Pcdhg* was indistinguishable from that of control WT cells at this time (Figure B', top graph), consistent with our observation that *Pcdhg* expression is not required for the migration of MGE-derived cINs. Strikingly, the survival fraction at 6 DAT of control *Pcdhg* WT (GFP+) and *Pcdhg* mutant (*Nkx2.1^Cre^*;Ai14 ;*Pcdhg^fcon3/fcon3^*) cINs at the injection site or at multiple locations anterior or posterior to the site of injection were also similar (*Figure 9B'*, bottom graph). By 21 DAT the survival of *Pcdhg* mutant (*Nkx2.1^Cre^*;Ai14 ;*Pcdhg^fcon3/fcon3^*) cells was dramatically reduced, and to a similar extent at all distances from the injection site (*Figure 9B and B'*). Since the density of cIN varies five-fold over regions measured, we conclude that the survival of control *Pcdhg* WT and *Pcdhg* mutant cIN does not depend on their density over this range.

In order to determine the absolute number of cINs eliminated in our co-transplantation experiments, we co-transplanted 50 K cells of each genotype (*Pcdhg* WT and *Pcdhg* mutant) into WT host mice (*Figure 10A*). Our baseline for survival was established at 6 DAT, before the period of cIN programmed cell death. In control experiments where cIN precursors WT for the *Pcdhg* allele, derived from *Nkx2.1^Cre^*;Ai14 or *Gad1-GFP* embryos, were transplanted, 39% of the transplanted cIN population was eliminated between 6 and 21 DAT (*Figure 10A–C* and *Figure 10—figure supplement 1*). Therefore, transplanted MGE cINs not only undergo programmed cell death during a period defined by their intrinsic cellular age but are also eliminated in a proportion that is strikingly similar to that observed during normal development (*Wong et al., 2018*; *Southwell et al., 2012*). Given these observations, we next asked how the presence of *Pcdhg* mutant cIN affected the survival of WT cIN in the transplantation setting. We co-transplanted 50K *Gad1*-GFP *Pcdhg* WT (GFP+) with 50K *Nkx2.1^Cre^*;Ai14 *Pcdhg* mutant (tdTomato+) MGE cIN precursors and compared the survival of each population at 6 and 21 DAT. At 6 DAT the total number of tdTomato+ cells in the cortex of recipient mice was similar to that of GFP+ cells (*Figure 10A,D and E*). However, between 6 and 21DAT, the total number of GFP+ cells had decreased by an average of ~63% (*Figure 10E*, compared to *Figure 10C*). Compared to the ~40% of endogenous or transplanted WT cINs that are normally eliminated (present study and previous work *Southwell et al., 2012*; *Wong et al., 2018*), this experiment suggests that WT cells die at a higher rate (63%) when co-transplanted with *Pcdhg* mutant MGE cells. However, this observation would require additional animals for statistical confirmation. Regardless, the number of *Pcdhg* mutant (tdTomato+ cells) cINs decreased dramatically, by ~96% (*Figure 10E*). This experiment confirms that MGE cells lacking *Pcdhg* function are eliminated in far greater numbers than control MGE cells and show that the presence of *Pcdhg* WT cINs within a

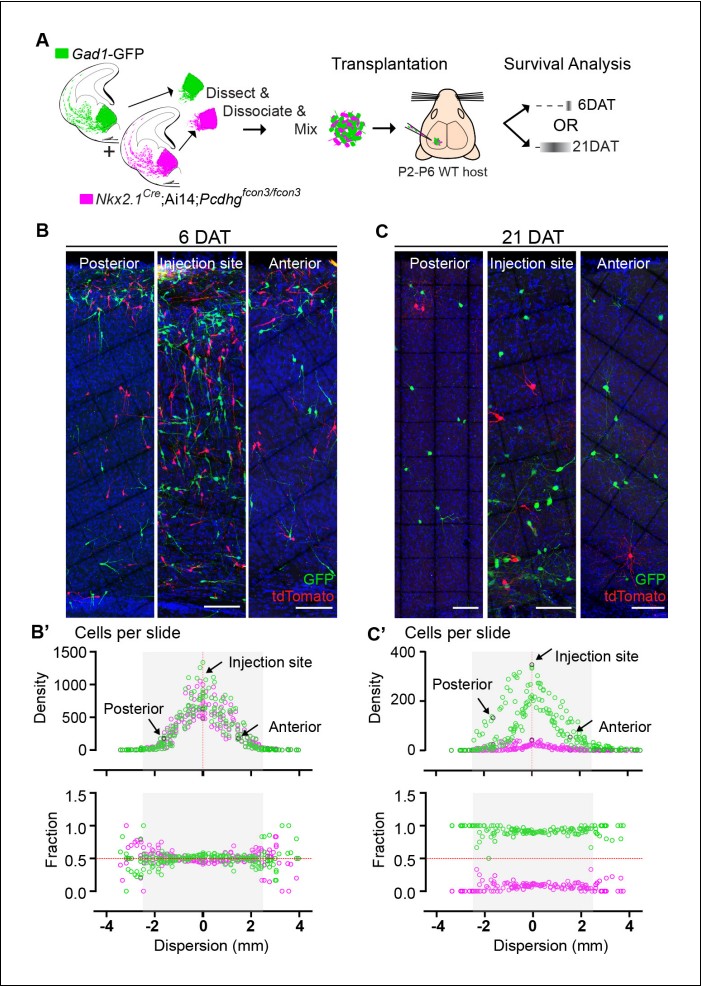

**Figure 9.** Survival of cIN, WT or mutant for *Pcdhg*, was not affected by cell density. (A) MGE cells derived from *Nkx2.1^Cre;*Ai14;*Pcdhg^fcon3/fcon3* embryos (*Pcdhg* mutant, magenta) and from *Gad1*-GFP embryos (*Pcdhg* WT, green) were mixed in equal numbers and transplanted into WT hosts. The survival of tdTomato and GFP- labeled cINs was analyzed in every other section throughout the brain region of the transplant dispersal. (B) Photographs of representative coronal sections at the injection site, or anterior and posterior to it, from host mice at 6DAT. Similar numbers of tdTomato and GFP-labeled cINs were observed at each location. Scale bar 100 μm. (B') Dispersion analysis at 6 DAT of the *Pcdhg* WT (green) or *Pcdhg* mutant (magenta) cells, represented as density (top) or survival fraction (bottom) as a function of distance from the site of injection in the host recipients. Note that the density of cells decreases as one moves anteriorly or posteriorly with respect to the injection site. At 6 DAT, the dispersal and survival was similar for both WT and *Pcdhg* mutant cells. (C) Photographs of representative coronal sections at the injection site, or anterior and posterior to it, from host mice at 21DAT. Note the dramatic reduction in the number of *Pcdhg* mutant cells (magenta) compared to the *Pcdhg* WT cells (green). Scale bar 100 μm. (C') Dispersion analysis at 21DAT of the *Pcdhg* WT (green) or *Pcdhg* mutant (magenta) cells, represented as density (top) or survival fraction (bottom) as a function of distance from the site of injection in the host recipients. At 21 DAT, the survival fraction for the *Pcdhg* mutant cells (magenta) was dramatically reduced and similarly affected at different locations with respect to the injection site.

The online version of this article includes the following source data for figure 9:

**Source data 1.** Dispersal and survival analysis of transplanted MGE-derived cIN precursor cells carrying WT or mutant *Pcdhg*.

mixed population also affects the survival of mutant cINs (compare *Figure 8* and *Figure 10*). These observations are consistent with the hypothesis that cINs interact with others of the same age discussed below.

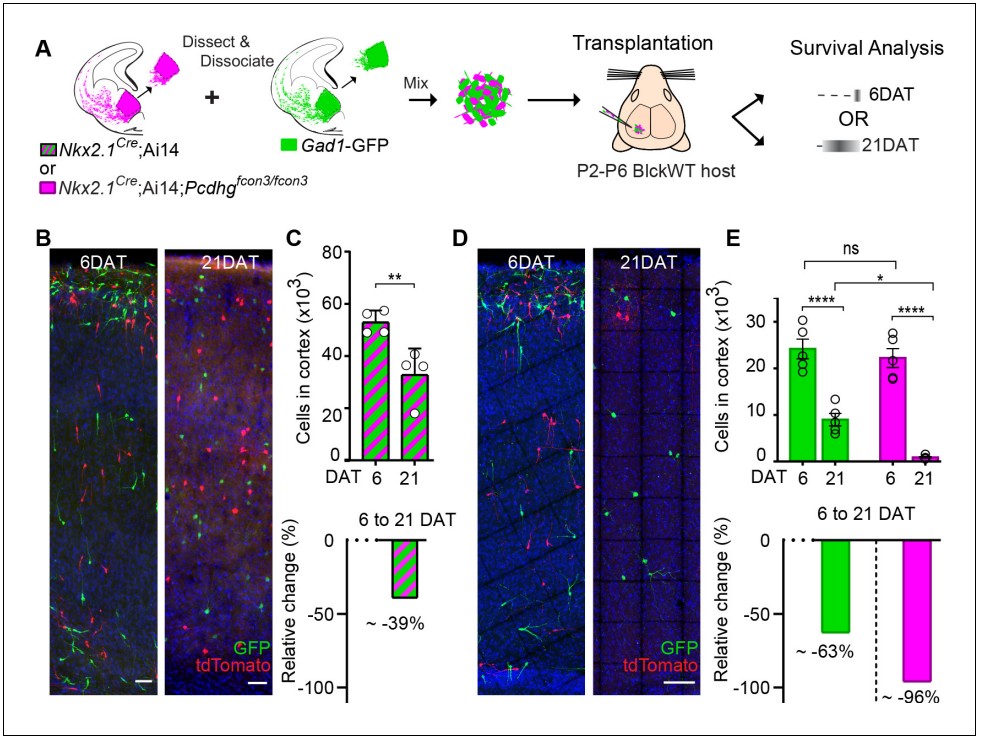

**Figure 10.** MGE cell transplantation reveals a non-cell autonomous effect of *Pcdhg* on cIN survival. (**A**) Schematic of co-transplantation experiment for quantification of absolute number of transplanted MGE cells derived from (1) *Nkx2.1^Cre*;Ai14 and *Gad1*-GFP embryos WT for *Pcdhg* or (2) *Nkx2.1^Cre*;Ai14;*Pcdhg^fcon3/fcon3* (*Pcdhg* mutant, magenta) and *Gad1*-GFP (*Pcdhg* WT, green) embryos. The total numbers of transplant-derived cINs were counted at 6 and 21 DAT throughout the volume of cortex were transplanted cells dispersed. (**B**) Photographs of representative coronal sections of co-transplanted tdTomato and GFP-labeled cells, both *Pcdhg* WT, at 6 and 21 DAT. Scale bar 100 um. (**C**) Absolute number of surviving tdTomato and GFP-labeled *Pcdhg* WT cIN at 6 and 21 DAT (top graph) (Mann-Whitney test, \*\*p=0.0286, n = 4 mice per time point from one transplant cohort). The drop in number of transplant derived cells was similar for WT-GFP+ and WT-tdTomato+ (*Figure 10—figure supplement 1*). A 39% drop in cIN number was observed between 6 and 21 DAT (bottom graph). (**D**) Photographs of representative coronal sections of transplanted tdTomato-labeled *Pcdhg* mutant (magenta) and GFP-labeled *Pcdhg* WT (green) cells at 6 and 21 DAT. Survival of the cINs drops for both genotypes, but the tdTomato-labeled cells were nearly eliminated by 21 DAT. Scale bar 100 μm. (**E**) Absolute number of surviving cINs at 6 and 21 DAT (top graph)(2-way ANOVA; $F_{age}$ = 128.65, P value < 0.0001, adjusted p value \*\*\*\*p<0.0001); $F_{genotype}$ = 9.74 (p value=0.0066, adjusted p value \*p=0.0126); n = 5 mice per time point from one transplant cohort). Comparing 6 and 21 DAT a drop of ~63% and of 96.0% was observed, respectively, for cells WT and mutant for *Pcdhg* (bottom graph).

The online version of this article includes the following source data and figure supplement(s) for figure 10:

**Source data 1.** Survival analysis and absolute quantification of transplanted MGE-derived cIN precursor cells carrying WT or mutant *Pcdhg*.

**Figure supplement 1.** Fluorescent reporter or breeding background does not affect survival of transplanted MGE-derived cINs.

**Figure supplement 1—source data 1.** Survival analysis and absolute quantification of transplanted MGE-derived cIN precursor cells carrying WT *Pcdhg* but carry different fluorophores.

## Loss of *Pcdhg* isoforms *Pcdhgc3*, *Pcdhgc4*, and *Pcdhgc5* is sufficient to increase cell death

The results above indicate that the loss of function of all 22 Pcdh isoforms encoded from the *Pcdhg gene* cluster significantly increased cell death among cINs. Whether all 22 *Pcdhg* are equally involved in the regulation of cIN survival remains unclear. Our qPCR expression analysis suggests that the expression of *Pcdhga1*, *Pcdha2*, *Pcdhgc4* and *Pcdhgc5* in cINs increases during, or soon after, the period of cell death (*Figure 1C*). To test if *Pcdhga1*, *Pcdhga2*, and *Pcdhga3* were required

for the normal survival of MGE-derived cINs, we crossed the $Pcdhg^{tako/tako}$ mouse line (*Pcdhga1, Pcdhga2, and Pcdhga3 isoform KO*) to *Nkx2.1^Cre^*; Ai14 mice (**Figure 7A**). At P30, the density of cINs in visual cortex of *Nkx2.1^Cre^;Ai14;Pcdhg^{tako/tako}^* (*Pcdhga1, Pcdhga2, and Pcdhga3 mutant*) mice was not significantly different from that of control *Nkx2.1^Cre^;Ai14* mice that are WT for *Pcdhg* (**Figure 7D**). Consistent with this finding, co-transplanted E13.5 *Nkx2.1^Cre^; Ai14;Pcdhg^{tako/tako}^* MGE cells and *Pcdhg* WT (GFP+) displayed surviving fractions of similar sizes (**Figure 11A,B & B'**). There-fore, the removal of the first three isoforms (*Pcdga1, Pcdhga2 and Pcdhga3*) of the *Pcdhg* cluster does not significantly affect cIN survival.

We next tested if removal of the last three isoforms of the *Pcdhg* cluster (*Pcdhgc3, Pcdhgc4, and Pcdhgc5*) affected cIN survival. *Pcdhg^{tcko/+}^* mice, heterozygous for excision of *Pcdhgc3, Pcdhgc4, and Pcdhgc5* were crossed to the *Nkx2.1^Cre^;Ai14* mouse line to label MGE/POA-derived cINs (**Figure 11A**). Even though homozygous *Nkx2.1^Cre^;Ai14 ;Pcdhg^{tcko/tcko}^* mice develop normally (nor-mal weight and no evidence of brain abnormalities) and are born in normal Mendelian ratios, these mice die shortly after birth (*Chen et al., 2012*). To bypass neonatal lethality, and study the role of *Pcdhgc3, Pcdhgc4, and Pcdhgc5* isoforms during the normal period of cIN programmed cell death, we co-transplanted *Pcdhg^{tcko/tcko}^* (tdTomato+, mutant) and *Pcdhg* WT (GFP+) E13.5 MGE cells into the cortex of WT neonatal recipients (**Figure 11A**). At 6 DAT, the dispersion and density of tdTo-mato+ and GFP+ cells were indistinguishable. However, the number of tdTomato+ MGE-derived *Pcdhg^{tcko/tcko}^* mutant cINs dropped dramatically between 6 and 21 DAT, compared to the *Pcdhg* WT (GFP+) population (**Figure 11C**). The survival of *Pcdhg^{tcko/tcko}^* mutant cIN at 21 DAT was strik-ingly similar to that observed after transplantation of MGE cells lacking the entire *Pcdhg* cluster (*Nkx2.1^Cre^;Ai14;Pcdhg^{fcon3/fcon3}^*); compare **Figure 8** and **Figure 11**. These results indicate that unlike *Pcdga1, Pcdhga2 and Pcdhga3* isoforms, *Pcdhgc3, Pcdhgc4, and Pcdhgc5* are essential for cIN survival.

## Morphological and physiological maturation of cINs lacking *Pcdhg*

The above results indicate that cINs lacking *Pcdhg* genes have increased cell death, specifically when the transplanted cells reach an age equivalent to that of endogenous cINs undergoing their normal period of programmed cell death. We therefore asked whether the loss of *Pcdhg* in cINs affected their morphological maturation during this period. We first determined the survival fraction for co-transplanted control *Gad1*-GFP (*Pcdhg* WT) and *Nkx2.1^Cre^;Ai14;Pcdhg^{fcon3/fcon3}^* (*Pcdhg* mutant) MGE-derived cIN precursors at two-day intervals during the intrinsic period of cIN cell death in the transplanted population (6, 8, 10 and 12 DAT). When equal proportions of *Pcdhg* WT and *Pcdhg* mutant cells were co-transplanted, their survival fraction remained similar up to 6 DAT, but the proportion of the *Pcdhg* mutant cells dropped steadily throughout the period of cell death (**Figure 12B**). Morphological reconstructions of the transplanted cells during this period of cIN pro-grammed cell death (**Figure 12A**) revealed no obvious differences between *the Pcdhg* mutant and control *Pcdhg* WT cells in neuritic complexity, including neurite length (**Figure 12C**), the number of neurites (**Figure 12D**), number of nodes (**Figure 12E**) and number of neurite ends (**Figure 12F**). These results suggest that *Pcdhg* genes do not play a major role in the morphological maturation of cINs during the period of cIN death.

Next, we utilized co-transplantation of cINs that were either *Pcdhg*-deficient (*Nkx2.1^Cre^;Ai14; Pcdhg^{fcon3/fcon3}^*) or WT (*Gad1*-GFP) to investigate whether the loss of *Pcdhg* affected the integration or intrinsic physiological properties of these cells at time points around the peak of *Pcdhg*-mediated cell death. To test how integration was affected, we made acute cortical slices of mouse visual cortex at 8, 9, 10, 11, and 12 DAT and measured the frequency of spontaneous excitatory (glutamatergic) and inhibitory (GABAergic) synaptic events, comparing mutant and WT cINs within the same slice. There was no effect of *Pcdhg* loss of function on the frequency of spontaneous excitatory (glutama-tergic) synaptic events or on the frequency of spontaneous inhibitory (GABAergic) synaptic events (**Figure 13B** and **Tables 1** and **2**). We next investigated whether the loss of *Pcdhg* function altered intrinsic physiological properties in co-transplanted cINs. There was no effect of the loss of *Pcdhg* on the maximum firing rate (**Figure 13C** and **Tables 1** and **2**), membrane time constant (Tau) (**Figure 13D** and **Table 1**), or input resistance (**Figure 13F** and **Table 1**). A difference in capacitance was observed between WT and *Nkx2.1^Cre^;Ai14 ;Pcdhg^{fcon3/fcon3}^* cINs at 8DAT, but this difference was not statistically significant following multiple comparisons correction and was not seen at later

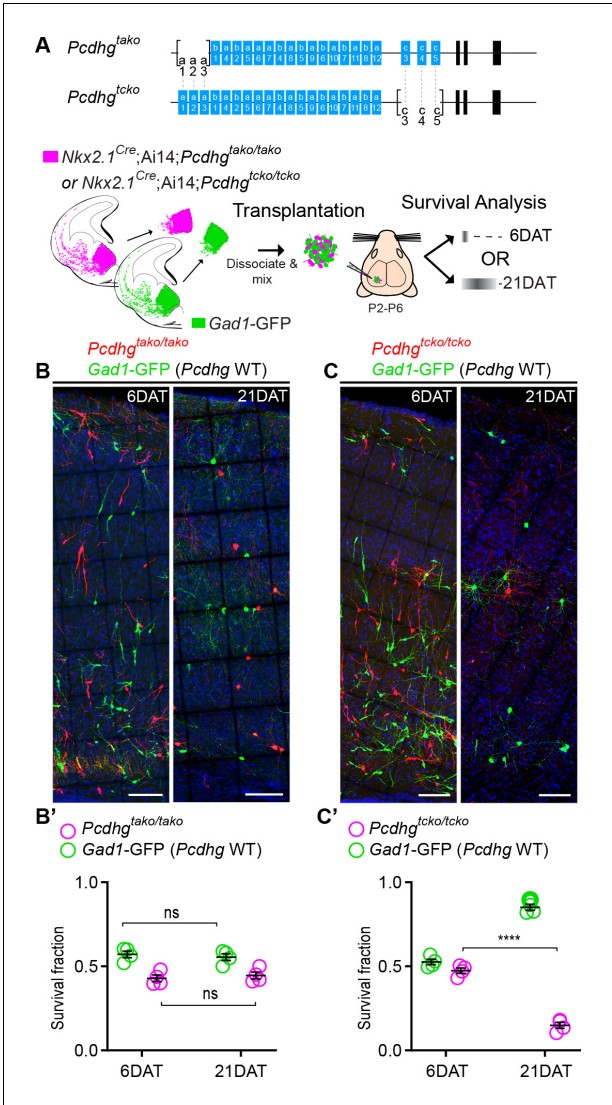

**Figure 11.** Loss of *Pcdhgc3*, *Pcdhgc4*, and *Pcdhgc5* is sufficient to increase cIN cell death. (**A**) Diagram of the mutant alleles *Pcdhg*^tako^ (*Pcdhga1*, *Pcdhga2*, and *Pcdhga3* KO) and *Pcdhg*^tcko^ (*Pcdhgc3*, *Pcdhgc4*, and *Pcdhgc5* KO). Below - schematic of transplantation of MGE cIN precursors from *Nkx2.1*^Cre^;Ai14;*Pcdhg*^tako/tako^ (*Pcdhga1*, *Pcdhga2*, and *Pcdhga3* deleted) and *Nkx2.1*^Cre^;Ai14;*Pcdhg*^tcko/tcko^ (*Pcdhgc3*, *Pcdhgc4*, and *Pcdhgc5* deleted) embryos. These cells were mixed in equal proportions with MGE cells from *Gad1*-GFP embryos (*Pcdhg* WT, green) and transplanted into WT Blk6 host recipients. Survival of the GFP and tdTomato-labeled cells was analyzed at 6 and 21 DAT. (**B, B'**) Representative photographs of cortical sections from transplanted host animals at 6 (left) and 21 (right) DAT. Note the similar proportions of *Pcdhg* WT (GFP+) and *Pcdhga1*, *Pcdhga2*, and *Pcdhga3* deleted cells (tdTomato+) at 6 and 21DAT. Scale bars, 100 µm. (**B'**) Quantifications of the survival fraction of the GFP (green) and tdTomato (magenta)-labeled MGE-derived cells at 6 and 21 DAT. Note, survival fraction remains similar and constant for both genotypes (*Pcdhg* WT and *Pcdhga1*, *Pcdhga2*, and *Pcdhga3* deleted cells) between 6 and 21 DAT (Mann-Whitney test, p=6571, n = 4 mice per time point from one transplant cohort). (**C, C'**) Representative photographs from coronal brain sections of transplanted host animals at 6 (left) and 21 (right) DAT. Scale bars, 100 µm. Survival of MGE-derived cINs from *Pcdhgc3*, *Pcdhgc4*, and *Pcdhgc5* deleted embryos (tdTomato+) is markedly different from MGE-derived cINs from *Pcdhg* WT embryos (GFP+). (**C'**) Survival fraction at 6 and 21 DAT of the *Pcdhg* WT (green) and *Pcdhgc3*, *Pcdhgc4*, and *Pcdhgc5* deleted cells (magenta) (Mann-Whitney test, ****p=0.0286, n = 4 mice per time point from one transplant cohort).

The online version of this article includes the following source data for figure 11:

**Source data 1.** Survival analysis of transplanted MGE-derived cIN precursor cells deficient in *Pcdha1*, *Pcdha2* and *Pcdha3* or deficient in Pcdhc3, Pcdhc4 and Pcdhc5.

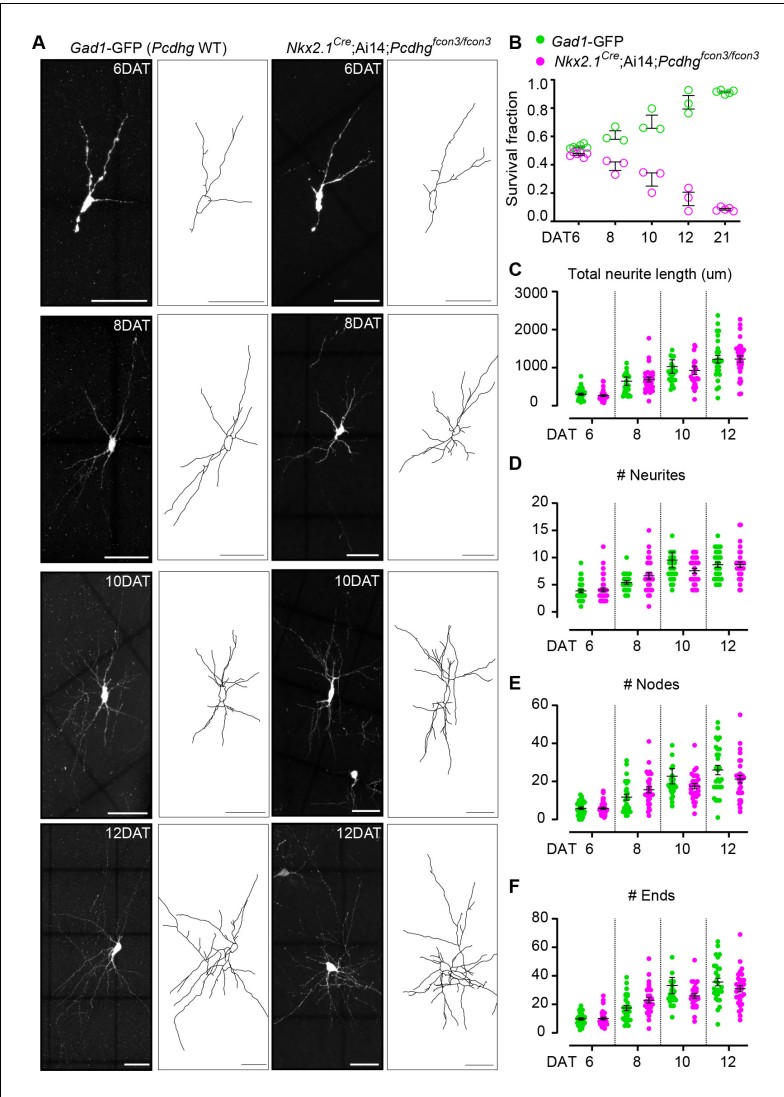

**Figure 12.** Loss of *Pcdhg* does not affect the morphological maturation of cIN during the period of programmed cell death. (A) Photographs of representative images and morphological reconstructions of co-transplanted *Gad1*-GFP cells (*Pcdhg* WT, left columns) with *Nkx2.1*^Cre;Ai14;*Pcdhg*^fcon3/fcon3 cells (*Pcdhg* mutant, right columns) at 6, 8, 10 and 12DAT. Scale bars, 50 μm. (B) Quantifications of *Pcdhg* WT(green) and *Pcdhg* mutant cells (magenta) from co-transplanted animals, represented as survival fraction from total number of cells per section at 6, 8 10 12 and 21 DAT. *Pcdhg* mutant cells begin to increase their elimination between 6 and 8 DAT and this increased death occurs through 21 DAT. (C–F) Measurements of neurite complexity during the period of programmed cell death, including neurite length (C), neurite number (D), node number (E) and neurite ends (F) in *Pcdhg* WT(green) and *Pcdhg* mutant (magenta) neurons at 6, 8, 10 and 12 DAT. Two-tailed unpaired *Student's t-test*, n = 32 (WT), n = 35 (*Pcdhg* mutant) cells at 6 DAT, n = 27 (WT and *Pcdhg* mutant) cells at 8 DAT, n = 26 (WT), n = 27 (*Pcdhg* mutant) cells at 10 DAT, and n = 27 (WT), n = 31 (*Pcdhg* mutant) cells at 12 DAT; cells analyzed from two transplant cohorts. All statistical comparisons were not significant following Benjamini-Hochberg multiple comparisons correction at alpha of 0.05.

The online version of this article includes the following source data for figure 12:

**Source data 1.** Morphological reconstruction of transplanted MGE-derived cIN precursor cells carrying WT or mutant *Pcdhg*.

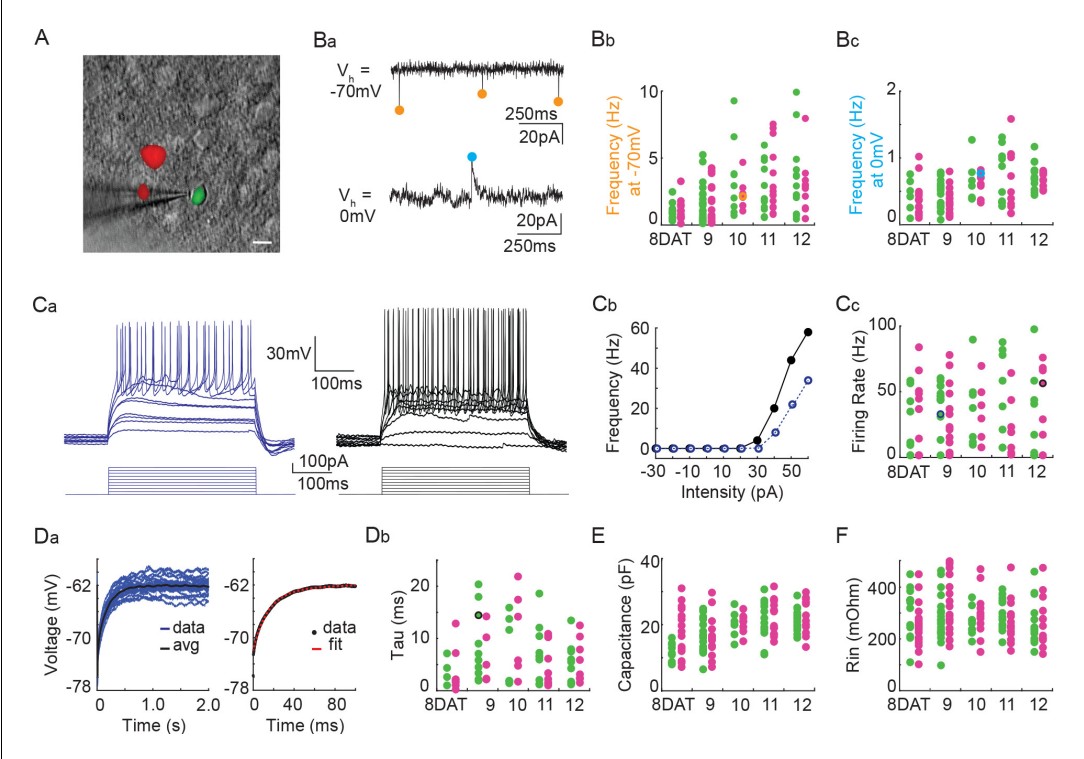

**Figure 13.** *Pcdhg* deletion does not affect the physiological properties of cINs during the period of programmed cell death. (**A**) DIC image with fluorescence image overlaid showing co-transplanted cINs from the MGE of *Gad1*-GFP (*Pcdhg* WT, green) or *Nkx2.1^Cre^*;Ai14;*Pcdhg^fcon3/fcon3^* (*Pcdhg* mutant, red) embryos, recorded in an acute brain slice taken from visual cortex (scale bar 10 µm). (**Ba**) Representative voltage clamp recordings (1 s) from a *Nkx2.1^Cre^*;Ai14;*Pcdhg^fcon3/fcon3^* (*Pcdhg* mutant) cINs held at −70 mV (top) to record glutamatergic events (orange circles) and 0 mV (bottom) to record GABAergic events (cyan circles). Bb, Bc. Group data from cINs recorded at 8, 9, 10, 11, and 12 DAT showing that co-transplanted *Gad1*-GFP cINs (WT, green circles) and *Nkx2.1^Cre^*;Ai14;*Pcdhg^fcon3/fcon3^* cINs (*Pcdhg* mutant, magenta circles) have similar rates of glutamatergic events (measured at −70 mV, **Bb**) and similar rates of GABAergic events (measured at 0 mV, **Bc**). The voltage clamp recordings from (**Ba**) are represented within the group data by the orange (−70 mV) and cyan (0 mV) circles in **Bb** and **Bc**) respectively. Ca, Cb. Representative current clamp traces showing a range of firing rates from a *Gad1*-GFP cIN (WT, blue trace) and a *Nkx2.1^Cre^*;Ai14;*Pcdhg^fcon3/fcon3^* (*Pcdhg* mutant, black trace) cIN responding to intracellular current injections (**Ca**), and the corresponding FI curves (**Cb**). (**Cc**) Group data from cINs recorded at 8, 9, 10, 11, and 12 DAT showing that co-transplanted *Gad1*-GFP cINs (WT, green circles) and *Nkx2.1^Cre^*;Ai14;*Pcdhg^fcon3/fcon3^* cINs (*Pcdhg* mutant, pink circles) have similar maximum spike rates. The current clamp traces from (**Ca** and **Cb**) are represented within the group data by the blue and black circles. (**Da**) Left: *Gad1*-GFP (WT) cIN voltage responses to repeated current injections (blue traces). Right: The membrane time constant (Tau) is calculated by fitting an exponential to the average voltage trace (black line). (**Db**) Group data from current clamp recordings of co-transplanted *Gad1*-GFP cINs (WT, green circles) and *Nkx2.1^Cre^*;Ai14;*Pcdhg^fcon3/fcon3^* cINs (*Pcdhg* mutant, magenta circles) at 8, 9, 10, 11, and 12 DAT shows that *Pcdhg* deletion does not affect membrane time constant. The current clamp recording from (**Da**) is represented within the group data by a black circle. (**E** and **F**) Group data from current clamp recordings of co-transplanted *Gad1*-GFP cINs (WT, green circles) and *Nkx2.1^Cre^*;Ai14;*Pcdhg^fcon3/fcon3^* cINs (*Pcdhg* mutant, magenta circles) at 8, 9, 10, 11, and 12 DAT shows that *Pcdhg* deletion does not affect either capacitance (**E**) or input resistance (**F**). Cells analyzed in **A**–**F** were taken from at least three transplant cohorts.

The online version of this article includes the following source data for figure 13:

**Source data 1.** Analysis of the intrinsic electrophysiological properties of transplanted MGE-derived cIN precursor cells carrying WT or mutant *Pcdhg*.

time points (*Figure 13E* and *Table 1*). We conclude that the synaptic integration and morphological - functional maturation of cINs lacking *Pcdhg* is similar to that of WT controls.

## Discussion

The findings above indicate that *Pcdhg* genes play a critical role in regulating cIN survival during the endogenous period of cIN programmed cell death. Specifically, *Pcdhgc3*, *Pcdhgc4*, and *Pcdhgc5* isoforms within the *Pcdhg* cluster are essential for the selection of those cINs that survive past the period of programmed cell death and become part of the adult cortical circuit. *Pcdhg* genes do not

affect the production or migration of cINs and appear to be dispensable for the survival of cINs after the period of cell death. Together with previous work in the spinal cord and retina, these results suggest that *Pcdhgc3*, *Pcdhgc4*, and *Pcdhgc5* isoforms are key to the regulation of programmed cell death in the CNS. In contrast, deletions of the alpha and beta *Pcdh* gene clusters did not alter cell death during this period.

Our initial approach involved the removal of *Pcdhg* function from all GAD2 expressing cells using the *Gad2^Cre*;Ai14;*Pcdhg^fcon3/fcon3* mice. These mice displayed a dramatic reduction of cortical interneurons of all subtypes, including a significant decrease in the number of VIP+ cells, which are derived from the CGE. A similar observation has been recently reported by *Carriere et al. (2020)*. In these mice, *Pcdhg* function was also removed from most other GABAergic neurons throughout the nervous system, as well as from a small fraction of astrocytes (*Taniguchi et al., 2011*). Since the removal of *Pcdhg* function in all GAD2-CRE expressing cells could affect the survival of cINs indirectly, we used *Nkx2.1^Cre*;Ai14 ; mice to more specifically remove *Pcdhg* function from MGE derived cINs. As in the *Gad2^Cre*;Ai14;*Pcdhg^fcon3/fcon3* mice, a sharp decrease in cINs was observed, but now only MGE-derived PV, SST and a subpopulation of RLN cIN were affected. The number of VIP cells was not affected in these mice, suggesting that the reduction in the number of MGE-derived cINs does not affect the survival of those derived from the CGE. Consistent with recent observations (*Carriere et al., 2020*), the number of un-recombined PV and SST (PV+/tdTomato- and SST+/tdTomato-) cINs in *Nkx2.1^Cre*;Ai14;*Pcdhg^fcon3/fcon3* mice, increased compared to WT mice (*Figure 3—figure supplement 4*). These cells, which are likely derived from the dorsal NKX6.2+/NKX2.1- MGE domain (*Hu et al., 2017a*; *Hu et al., 2017b*; *Fogarty et al., 2007*; *Sousa et al., 2009*) may increase their survival in compensation for the loss of Nkx2.1-derived cINs lacking *Pcdhg* function. However, we cannot exclude that the increased number of un-recombined PV and SST cells in *Nkx2.1^Cre*;Ai14; *Pcdhg^fcon3/fcon3* mice resulted from increased production or migration of cINs derived from regions of low, or no, expression of Nkx2.1. Further experiments will be required to understand the origin of these un-recombined PV+/Ai14- and SST+/Ai14- cINs and whether the observed increase in their numbers is due to compensatory survival mechanisms.

*Pcdhg* genes are not required for the normal production of young Nkx2.1-derived cINs in the MGE, or for their migration into the mouse cerebral cortex. The extent of proliferation in the MGE was essentially the same in *Pcdhg* WT and mutant mice, and the number of migrating MGE-derived cINs into the cortex was also indistinguishable between control and mutant mice. We did not directly address whether interference with the *Pcdha* and *Pcdhb* gene clusters affected the birth and migration of cINs, but we infer these two clusters also have no, or minimal effects on cIN production and migration because the final numbers of MGE-derived cINs were not significantly affected after the loss of either *Pcdha* or *Pcdhb*. However, the aggregate loss of *Pcdh* genes in multiple clusters might in principle be required for phenotypes to be manifested (*Ing-Esteves et al., 2018*). Our findings, therefore, cannot exclude the possibility that simultaneous elimination of the *Pcdha* and *Pcdhb* isoforms might have an effect on cIN production, migration, or apoptosis. However, the elimination of *Pcdhg* alone, and specifically of the *Pcdhgc3*, *Pcdhgc4*, and *Pcdhgc5* isoforms increases cell death among cINs, establishing that removal of a limited number of isoforms in the *Pcdhg* cluster is sufficient to reveal the cell-death phenotype.

Importantly, the increase in programmed cell death observed following the loss of *Pcdhg* function was fully rescued when the pre-apoptotic gene *Bax* was also eliminated. Not only was the increased *Pcdhg*-dependent cell death eliminated in *Bax* mutant animals, but these animals had ~40% increase in the numbers of surviving cINs compared to WT controls, identical to the effect of the *Bax* mutation in wild type animals. This observation is consistent with previous observation showing that ~ 40% of cINs are eliminated during the period of programmed cell death (*Southwell et al., 2012*; *Wong et al., 2018*). Moreover, the increased death of cINs after removal of *Pcdhg* function occurs precisely during the normal period of programmed cell death. These observations indicate that *Pcdhg* isoforms are required specifically to regulate cIN numbers during the critical window of programmed cell death. This is consistent with previous studies in the retina and spinal cord that have pointed to *Pcdhg*, and specifically the C-isoforms (*Pcdhgc3, Pcdhgc4 and Pcdhgc5*), as key mediators of programmed cell death (*Lefebvre et al., 2008*; *Chen et al., 2012*). Interestingly, in all three neural structures, the cortex, the spinal cord, and the retina, *Pcdhg* C-isoforms appear to be the key regulators of survival of local circuit interneurons.

A recent study suggests that the *Pcdhgc4* isoform is the key mediator in the regulation of neuronal cell death in the spinal cord (*Garrett et al., 2019*). How the specific *Pcdhgc4* isoform in the *Pcdhg* gene cluster mediates cell death remains a fundamental question for future research. Interestingly, the *Pcdhgc4* isoform appears to be unique in that it is the only *Pcdh* isoform that does not bind in a homophilic manner (*Garrett et al., 2019*) and it is not translocated to the membrane unless it is associated with other *Pcdha* or *Pcdhg* (*Aye et al., 2014*). The mechanism by which the *Pcdhgc4* isoform regulates cell-cell interactions among young cINs, leading to the adjustment of local circuit neuron numbers, remains unclear.

Heterochronic transplantation of cIN from the MGE into the cortex of WT mice allowed us to test for the survival of *Pcdhg* WT and *Pcdhg* mutant cIN simultaneously and in the same environment. As previously reported (*Southwell et al., 2012*), cINs die following their own time-course of maturation. Consistent with this, the transplanted cells (extracted from the MGE at E13.5) died with a delay of 6–12 days compared to the endogenous host cINs when transplanted into P0-P6 WT mice. This is consistent with the notion that the cellular age of cIN determines the timing of programmed cell death (*Southwell et al., 2012*). Interestingly, the role of *Pcdhg* function was clearly evident in these co-transplants in that the survival of the mutant cells was extremely low compared to that of WT cells. The dramatic decrease in the survival of *Pcdhg* mutant cINs occurs precisely during the period of programmed cell death for the transplanted population. The survival of transplanted WT cINs as well as that of *Pcdhg* mutant cINs was constant over a wide range of densities, as evidenced by the fact that while the density of transplanted cINs decreases as a function of the distance from the transplantation site, the proportion of dying cells of both phenotypes was similar at different distances from the site of transplantation.

Interestingly, the survival of WT cIN may also be reduced when co-transplanted with *Pcdhg* deficient MGE-cells although this difference did not reach statistical significance with the numbers of cases studied. If true, these findings would be consistent with the notion that cell-cell interactions among young cIN after their migration is an essential step in determining their final numbers. However, we cannot exclude that the increase in the elimination of WT cells may result from a non-specific (e.g., toxic) effect of the increased cell death among *Pcdhg* mutant cells. If the latter occurs, the process is specific to the population of Nkx2.1$^+$ MGE-derived cINs because there was no effect on cell death of WT CGE-derived VIP cINs (*Figure 3E*, *Figure 3—figure supplement 1*) or on non-Nkx2.1-derived SST or PV cells (*Figure 3—figure supplement 4*). Interactions mediated by Pcdhg, and specifically among the C-isoforms, may directly or indirectly regulate survival of cINs of the same age and origin.

Unlike the spinal cord where cell death takes place prenatally (*Wang et al., 2002b*; *Prasad et al., 2008*), cIN programmed cell death occurs mostly postnatally. Since mice lacking *Pcdhgc3*, *Pcdhgc4*, and *Pcdhgc5* isoforms die soon after birth, we could not study normal cIN cell death directly in these mutant animals. We, therefore, took advantage of transplantation and co-transplantation to compare the survival of cells lacking these three isoforms. The loss of cINs lacking *Pcdhgc3*, *Pcdhgc4*, and *Pcdhgc5* isoforms was identical to that when the function of the entire *Pcdhg* cluster is lost. This further suggests that these *Pcdhg* C-isoforms (*Pcdhgc3*, *Pcdhgc4*, and *Pcdhgc5*) are the key to the regulation of cIN death. The co-transplantation assay, implemented in the present study, provides strong evidence that *Pcdhg* in cINs are key to their selection by programmed cell death. *Pcdhg* could be mediating initial cell-cell interactions that are important for the survival of cINs. Two non-exclusive possibilities exist: (1) *Pcdhg* mediate cell-cell interaction among young cINs to adjust their population size, and levels of inhibition, according to the numbers that reached the cortex; (2) *Pcdhg* mediate interactions with locally produced excitatory pyramidal neurons to adjust final numbers according to local levels of excitation. For the latter, MGE-derived cIN could interact with pyramidal neurons via *Pcdhg* C-isoforms. However, alternative # 2 is unlikely to explain how *Pcdhg* adjust cIN numbers since using conditional removal of *Pcdhg* in pyramidal cells shows no effect on the survival of cINs (*Carriere et al., 2020*). However, we cannot exclude that initial connectivity with excitatory pyramidal neurons may indeed require the proper expression of *Pcdhg* among cINs through non-homophilic interactions.

A recent study has shown that coordinated activity of synaptically connected assemblies of cINs is essential for their survival (*Duan et al., 2020*). Pyramidal cells receive information from these assemblies via GABA$_A$γ2-signaling and through the de-synchronization of their activity regulate cIN programmed cell death. *Pcdhg* could be important in bringing together cINs of a common origin and at

similar stages of maturation for the formation of initial cIN functional assemblies. The formation of these assemblies of synchronously firing cINs and the subsequent selection by pyramidal driven de-synchronization could explain both cell/population autonomous (*Southwell et al., 2012*), and non-cell autonomous (*Wong et al., 2018*) mechanisms of cIN programmed cell death. Interestingly, PCDHGC5 binds to the GABA$_A\gamma$ subunit of the GABA receptor (*Li et al., 2012*), but the role of PCDHGC5-GABA$_A\gamma$ interaction on neuronal survival remains unknown. The transplantation assay provides a powerful tool to further study how *Pcdhg*, cell-cell interactions, and cellular age contribute to cIN selection. It will be interesting, for example, to determine if heterochronically transplanted cINs form functional assemblies and whether these assemblies are affected by the removal of different *Pcdhg* isoforms.

During the evolution of multiple mammalian species including that of humans, the cerebral cortex has greatly expanded in size and in the number of excitatory and inhibitory neurons it contains. Interestingly, the proportion of cINs to excitatory pyramidal neurons has remained relatively constant. Appropriate numbers of inhibitory cINs are considered essential in the modulation of cortical function. The embryonic origin of cINs, far from the cerebral cortex, raises basic questions about how their numbers are ultimately controlled in development and during evolution. Coordinated increased production of inhibitory interneurons in the MGE and CGE is an essential step to satisfy the demand of an expanded cortex (*Hansen et al., 2013*). In addition, MGE and CGE derived interneurons in larger brains require longer and more protracted migratory periods (*Paredes et al., 2016*). Interneurons arrive in excess of their final number. This is ultimately adjusted by a period of programmed cell death once the young cINs have arrived in the cortex. Here we have identified the C-isoforms (*Pcdhgc3, Pcdhgc4 and Pcdhgc5)* in the *Pcdhg* cluster as an essential molecular component that regulates programmed cell death among cINs. The fact that a cell surface adhesion protein plays a key role in this regulation suggests that interactions with other cells, possibly other cINs of the same age (*Southwell et al., 2012*), or possibly excitatory pyramidal cells (*Wong et al., 2018*), is part of the logic to adjust the final number of these essential GABAergic cells for proper brain function. An understanding of the cell-cell interactions that use *Pcdhg* C-isoform to regulate cIN cell death should give fundamental insights into how the cerebral cortex forms and evolves.

# Materials and methods

**Key resources table**

| Reagent type (species) or resource | Designation | Source or reference | Identifiers | Additional information |
|---|---|---|---|---|
| Genetic reagent (*mouse*) | *Gad2$^{Cre}$* | PMID:21943598 | | Also referred to as *Gad2*-IRES-Cre knock-in |
| Genetic reagent (*mouse*) | *Nkx2.1$^{Cre}$* | PMID:17990269 | | Also referred to as C57BL/6J-Tg(*Nkx2-1*-cre)2Sand/J |
| Genetic reagent (*mouse*) | *PV$^{Cre}$* | PMID:15836427 | | Also referred to as B6;129P2-*Pvalbtm1(cre)Arbr*/J |
| Genetic reagent (*mouse*) | *Sst$^{Cre}$* | PMID:21943598 | | Also referred to as *Ssttm2.1(cre)Zjh*/J |
| Genetic reagent (*mouse*) | Ai14 | The Jackson Laboratory | | Also referred to as Ai14 , Ai14 *D or* Ai14 (*RCL-tdT*)-*D* |
| Genetic reagent (*mouse*) | *Gad1*-GFP | The Jackson Laboratory | | Also referred to as *G42* line. |
| Genetic reagent (*mouse*) | *Bax$^{-/-}$* | The Jackson Laboratory | | Also referred to as B6;129-Baxtm2Sjk Bak1tm1Thsn/J |

*Continued on next page*

*Continued*

| Reagent type (species) or resource | Designation | Source or reference | Identifiers | Additional information |
|---|---|---|---|---|
| Genetic reagent (*mouse*) | *Pcdha*$^{acon/acon}$ | PMID:28450636 | | Referred to as *Pcdhα*$^{f/f}$ and *Pcdhα*$^{-/-}$ in original publication. |
| Genetic reagent (*mouse*) | *Pcdhb*$^{del/del}$ | PMID:28450637 | | Referred to as *Pcdhβ*$^{-/-}$ in original publication. |
| Genetic reagent (*mouse*) | *Pcdhg*$^{tako/tako}$; *Pcdhga1, Pcdhga2, and Pcdhga3* mutant; *Pcdhga1, Pcdhga2, and Pcdhga3* KO | PMID:22884324 | | Referred to as *Pcdhg*$^{tako/tako}$ in original publication. |
| Genetic reagent (*mouse*) | *Pcdhg*$^{tcko/tcko}$; *Pcdhgc3, Pcdhgc4, and Pcdhgc5* mutant; *Pcdhgc3, Pcdhgc4, and Pcdhgc5* KO | PMID:22884324 | | Referred to as *Pcdhg*$^{tcko/tcko}$ in original publication. |
| Genetic reagent (*mouse*) | *Pcdhg*$^{fcon3/fcon3}$ | PMID:19029044 | | Referred to as *Pcdh-γ*$^{fcon3}$ in original publication. |
| Antibody | anti-GFP (chicken polyclonal) | Aves Lab | Cat# GFP-1020, RRID:AB_10000240 | IF(1:2500) |
| Antibody | Anti-Reelin (mouse monoclonal) | MBL International | Cat#: MBL, D223–3, RRID:AB_843523 | IF(1:500) |
| Antibody | Anti-PV (rabbit antiserum) | Swant | Cat#: PV27 , RRID:AB_2631173 | IF(1:1000) |
| Antibody | Anti-PV (mouse monoclonal) | Sigma-Aldrich | Cat#: P3088, RRID:AB_477329 | IF(1:500) |
| Antibody | Anti-SST (rat, polyclonal | Santa Cruz Biotechnology | Cat#:sc-7819, RRID:AB_2302603 | IF(1:500) |
| Antibody | Anti-cleaved caspase 3 (rabbit polyclonal) | Cell Signaling Technology | Cat#: 9661L, RRID:AB_2341188 | IF(1:400) |
| Antibody | Anti-phosphohiston-H3 (rabbit polyclonal) | EDM Millipore | Cat#: 06–570, RRID:AB_310177 | IF(1:500) |
| Antibody | Anti-NKX2-1 (rabbit polyclonal) | Life Technologies | Cat#: sc-13040, ARRID:AB_793532 | IF(1:250) |
| Chemical compound, drug | DNAse I | Sigma Millipore | Cat#: 260913-10MU | 180 ug/mL |
| Commercial assay or kit | QuantiTect Rev. Transcription Kit | Qiagen | Cat#: 205311 | |
| Software, algorithm | Stereo Investigator | MBF bioscience | | |
| Software, algorithm | Neurolucida | MBF bioscience | | |

*Continued on next page*

Continued

| Reagent type (species) or resource | Designation | Source or reference | Identifiers | Additional information |
|---|---|---|---|---|
| Software, algorithm | custom software written in MATLAB | 'other' | | *Larimer, 2020*. mPhys. MATLAB Central File Exchange. (https://www.mathworks.com/matlabcentral/fileexchange/21903-mphys). 1.2.0.0. |

## Animals

R26-Ai14 , *Gad1*-GFP, *Gad2*-ires-Cre (*Gad2^Cre^*), BAC-*Nkx2.1*-Cre (*Nkx2.1^Cre^*), *Sst*-ires-Cre (*Sst^Cre^*), PV-IRES-Cre-pA (*PV^Cre^*), *Bax^-/-^*, *Bax^fl/fl^* and WT C57BL/6J breeders were purchased from the Jackson Laboratory. Whenever possible, the number of males and females was matched for each experimental condition. All protocols and procedures followed the University of California, San Francisco (UCSF) guidelines and were approved by the UCSF Institutional Animal Care Committee.

*Pcdhg* loss of function mice were obtained by crossing *Pcdhg^fcon3/fcon3^* mice with *Gad2^Cre^*;Ai14; *Pcdhg^fcon3/+^* mice, *Nkx2.1^Cre^*;Ai14;*Pcdhg^fcon3/+^*, *PV^Cre^*;Ai14;*Pcdhg^fcon3/+^* or *Sst^Cre^*;Ai14;*Pcdhg^fcon3/+^*. *Pcdhga1, Pcdhga2 and Pcdhga3* isoform knockout mice were obtained from crosses of *Pcdhg^tako/tako^* mice to *Nkx2.1^Cre^*;Ai14;*Pcdhg^tako/+^* mice.

*Pcdha* loss function mice were obtained by crossing *Pcdha^acon/acon^* mice with *Nkx2.1^Cre^*;Ai14; *Pcdha^acon/+^* mice. *Pcdhb* loss function mice were obtained by crossing *Pcdhb^del/del^* mice with *Nkx2.1^Cre^*;Ai14;*Pcdhb^del/+^* mice. Embryonic donor tissue was produced by crossing WT C57BL/6J to heterozygous mice expressing green fluorescent protein-expressing (GFP) driven by *Gad1 promoter*. *Pcdhg^fcon3/fcon3^* (tdTomato-expressing) tissue was obtained from embryos produced by crossing *Nkx2.1^Cre^*;Ai14;*Pcdhg^fcon3/+^* mice with *Pcdhg^fcon3/fcon3^* mice. *Pcdhg^tako/tako^* (tdTomato-expressing) tissue was obtained from embryos produced by crossing *Nkx2.1^Cre^*;Ai14;*Pcdhg^tako/+^* mice to homozygote *Pcdhg^tako/tako^* mice. *Pcdhg^tcko/tcko^* (tdTomato-expressing) tissue was obtained from embryos produced by crossing *Nkx2.1^Cre^*;Ai14;*Pcdhg^tcko/+^* mice to heterozygote *Pcdhg^tcko/+^* mice. Homozygous *Pcdhg^tcko/tcko^* mice die around birth. GAD1-GFP and *Nkx2.1^Cre^*; Ai14 offspring were genotyped under an epifluorescence microscope (Leica), and PCR genotyping was used to screen for *Pcdhg^fcon3/fcon3^*, *Pcdhg^tako/tako^* and *Pcdhg^tcko/tcko^* embryos in GFP or Ai14 positive animals. All cell transplantation experiments were performed using wild type C57Bl/6 recipient mice. All mice were housed under identical conditions.

## Immunostaining

Mice were fixed by transcardiac perfusion with 10 mL of ice-cold PBS followed by 10 mL of 4% formaldehyde/PBS solution. Brains were incubated overnight (12–24 hr) for postfixation at 4°C, then rinsed with PBS and cryoprotected in 30% sucrose/PBS solution for 48 hr at 4°C. Unless otherwise stated, immunohistochemistry was performed on 50 µm floating sections in Tris Buffered Saline (TBS) solution containing 10% normal donkey serum, 0.5% Triton X-100 for all procedures on postnatal mice. Immunohistochemistry from embryonic tissue was performed on 20 µm cryostat sections. All washing steps were done in 0.1% Triton X-100 TBS for all procedures. Sections were incubated overnight at 4°C with selected antibodies, followed by incubation at 4°C overnight in donkey secondary antibodies (Jackson ImmunoResearch Laboratories). For cell counting and *post hoc* examination of marker expression, sections were stained using chicken anti-GFP (1:2500, Aves Labs, GFP-1020, RRID:AB_10000240), mouse anti-Reelin (1:500 MBL, D223–3, RRID:AB_843523), rabbit anti-PV (1:1000, Swant PV27 , RRID:AB_2631173), mouse anti-parvalbumin (anti-PV, 1:500, Sigma-Aldrich, P3088, RRID:AB_477329), rat anti-somatostatin (SST, 1:500, Santa Cruz Biotechnology, sc-7819, RRID:AB_2302603), anti-cleaved caspase 3 (1:400, Cell Signaling Technology, 9661L, RRID:AB_2341188), rabbit anti-phosphohistone-H3 (1:500; EDM Millipore, RRID:AB_310177), rabbit anti-NKX2-1 (1:250, Life Technologies, RRID:AB_793532).

## Cell counting

For cell density counts in the visual and barrel cortex (*Figure 2C & C'*, *Figure 3C & C'*, *Figure 5C*, *Figure 6* and *Figure 7*), cells were directly counted using a Zeiss Axiover-200 inverted microscope (Zeiss) and an AxioCam MRm camera (Zeiss), using Stereo Investigator (MBF). tdTomato+ cells were counted in every six sections (i.e., 300 μm apart) along the rostral-caudal axis of visual and barrel cortex. Total cell counts were extrapolated by Stereo Investigator. Cell densities were determined by dividing the total number of tdTomato+ cells by the volume of the region of interest in the visual or barrel cortex, identified by landmarks, for each animal. To measure PV, SST, RLN and VIP-positive cell densities in the visual cortex (*Figure 2D* and *Figure 3E*), cells were counted from confocal-acquired images. Cleaved caspase-3-positive (CC3+) cells were counted from images acquired on a Zeiss Axiover-200 inverted microscope (Zeiss) and an AxioCam MRm camera (Zeiss) using Neurolucida (MBF). Cleaved caspase-3-positive cells were counted in the cortex of every six sections along the rostral-caudal axis for each animal (Figure A and B). Cells in the olfactory bulb, hippocampus, and piriform cortex were not counted. For layer distribution analysis of tdTomato+ cells and for analysis of tdTomato negative (non-recombined) PV and SST positive cells (*Figure 2—figure supplement 2*, *Figure 3—figure supplements 3* and *4*, cells were counted from confocal-acquired images and using Fiji software cell counter function.

## Cell dissection and transplantation

Unless otherwise mentioned, MGEs were dissected from E13.5 embryos as previously described (*Southwell et al., 2012*). The day when the sperm plug was observed was considered E0.5. Dissections were performed in ice-cold Leibovitz L-15 medium. MGEs were kept in L-15 medium at 4℃. MGEs were mechanically dissociated into a single cell suspension by repeated pipetting in L-15 medium containing DNAse I (180 ug/ml). The dissociated cells were then concentrated by centrifugation (4 min, 800xg). For all co-transplantations, the number of cells in each suspension (GFP+ or tdTomato+) was determined using a hemocytometer. Concentrated cell suspensions were loaded into beveled glass micropipettes (≈70–90 μm diameter, Wiretrol 5 μl, Drummond Scientific Company) prefilled with mineral oil and mounted on a microinjector. Recipient mice (C57Bl/6) were anesthetized by hypothermia (~4 min) and positioned in a clay head mold that stabilizes the skull (*Merkle et al., 2007*). Micropipettes were positioned at an angle of 0 degrees from vertical in a stereotactic injection apparatus. Unless otherwise stated, injections were performed in the left hemisphere 1 mm lateral and 1.5 mm anterior from Lambda, and at a depth of 0.8 mm from the surface of the skin. After the injections were completed, transplant recipients were placed on a warm surface to recover from hypothermia. The mice were then returned to their mothers until they were perfused or weaned (P21). Transplantation of *Nkx2.1$^{Cre}$*;Ai14;*Pcdhg$^{tako/tako}$* was performed using frozen cells (*Figure 11B*). For these experiments, dissected MGEs from each embryo were collected in 500 uL L15 and kept on ice until cryopreserved. MGEs were resuspended in 10% DMSO in L15 and cryopreserved as previously described (*Rodríguez-Martínez et al., 2017*). Vials were cooled to −80℃ at a rate of −1 ℃/minute in a Nalgene Mr. Frosty Freezing Container and then transferred to liquid nitrogen for long term storage. Prior to transplantation, vials were removed from −80℃ and thawed at 37℃ for 5 min, after which the freezing medium was removed from the vial and replaced with 37 ℃ L-15. Dissociation was performed as above.

For cell counts from transplanted animals, *Gad1*-GFP positive cells and tdTomato-positive cells were counted in all layers of the neocortex. Cells that did not display neuronal morphology were excluded from all quantifications. The vast majority of cells transplanted from the E13.5 MGE exhibited neuronal morphologies in the recipient brain. GFP and tdTomato-positive cells were counted from tiles acquired on a Zeiss Axiover-200 inverted microscope (Zeiss) with an AxioCam MRm camera (Zeiss); using Neurolucida (MBF). For quantification of the absolute numbers of transplanted cells in the neocortex of host recipients, cells from every second coronal section were counted (*Figure 9*). The raw cell counts were then multiplied by the inverse of the section sampling frequency (2) to obtain an estimate of total cell number (*Figure 10*). In some experiments the absolute number of grafted cells varies between transplants; hence we report findings as the fraction of cells (GFP or tdTomato) that survive from the total transplant-derived cell number in that animal (GFP + tdTomato-positive) (*Figure 8*). We determined for these experiments the number of transplant-derived cells of the different genotypes before and, in different animals, after the period of cell death. For

**Table 1.** Group data for the frequency of spontaneous excitatory and inhibitory synaptic currents (Hz), max firing rate (Hz), tau (ms), capacitance (pF), and input resistance (mOhm) from *Figure 13*.
The mean, standard deviation, sample size, and statistical tests are reported. Comparisons were not statistically significant following Benjamini-Hochberg multiple comparisons correction at an alpha of 0.05.

| | Frequency of spontaneous excitatory synaptic currents (Hz) | | | Frequency of spontaneous inhibitory synaptic currents (Hz) | | |
|---|---|---|---|---|---|---|
| | $Pcdhg^{fcon3/fcon3}$ | WT | Mann-Whitney | $Pcdhg^{fcon3/fcon3}$ | WT | Mann-Whitney |
| | Mean ± SD | Mean ± SD | p value | Mean ± SD | Mean ± SD | p value |
| 8DAT | 1.0 ± 0.7 (n = 18) | 1.0 ± 0.7 (n = 9) | 0.890 | 0.4 ± 0.2 (n = 18) | 0.4 ± 0.2 (n = 6) | 0.463 |
| 9DAT | 1.4 ± 1.4 (n = 19) | 1.6 ± 1.4 (n = 21) | 0.432 | 0.4 ± 0.2 (n = 15) | 0.4 ± 0.2 (n = 18) | 0.347 |
| 10DAT | 2.2 ± 1.0 (n = 9) | 3.4 ± 2.8 (n = 9) | 0.652 | 0.6 ± 0.2 (n = 9) | 0.6 ± 0.3 (n = 7) | 0.859 |
| 11DAT | 3.2 ± 2.3 (n = 14) | 2.5 ± 1.8 (n = 13) | 0.298 | 0.6 ± 0.5(n = 11) | 0.7 ± 0.4 (n = 11) | 0.711 |
| 12DAT | 2.6 ± 2.1 (n = 11) | 3.3 ± 2.8 (n = 14) | 0.597 | 0.7 ± 0.1 (n = 10) | 0.7 ± 0.2 (n = 13) | 0.436 |
| | Max Firing Rate (Hz) | | | Tau (ms) | | |
| | $Pcdhg^{fcon3/fcon3}$ | WT | Mann-Whitney | $Pcdhg^{fcon3/fcon3}$ | WT | Mann-Whitney |
| | Mean ± SD | Mean ± SD | p value | Mean ± SD | Mean ± SD | p value |
| 8DAT | 21.8 ± 23 (n = 16) | 21.5 ± 22.9 (n = 12) | 0.789 | 3.6 ± 4.3 (n = 8) | 3.8 ± 2.6 (n = 4) | 0.683 |
| 9DAT | 29.1 ± 25.1 (n = 17) | 35.0 ± 22.3 (n = 19) | 0.416 | 6.8 ± 5.3 (n = 5) | 8.7 ± 6.7 (n = 10) | 0.717 |
| 10DAT | 35.7 ± 22.8 (n = 6) | 37.7 ± 28.6 (n = 7) | 1.000 | 11.0 ± 8 (n = 6) | 7.8 ± 6.7 (n = 6) | 0.387 |
| 11DAT | 24.5 ± 24.4 (n = 11) | 36.3 ± 34.9 (n = 12) | 0.534 | 5.7 ± 4.3 (n = 10) | 7.3 ± 5.7 (n = 9) | 0.707 |
| 12DAT | 35.0 ± 32.3 (n = 10) | 29.8 ± 29.3 (n = 12) | 0.757 | 5.8 ± 4.1 (n = 8) | 5.5 ± 3.9 (n = 9) | 0.880 |
| | Capacitance (pF) | | | Input Resistance (mOhm) | | |
| | $Pcdhg^{fcon3/fcon3}$ | WT | Mann-Whitney | $Pcdhg^{fcon3/fcon3}$ | WT | Mann-Whitney |
| | Mean ± SD | Mean ± SD | p value | Mean ± SD | Mean ± SD | p value |
| 8DAT | 18.3 ± 7.3 (n = 16) | 11.6 ± 2.7 (n = 10) | 0.019 | 257.3 ± 87.1 (n = 23) | 313.6 ± 86.5 (n = 16) | 0.027 |
| 9DAT | 16.4 ± 6.0 (n = 15) | 17.2 ± 5.1 (n = 19) | 0.465 | 327 ± 106.5 (n = 23) | 278.9 ± 83.4 (n = 23) | 0.095 |
| 10DAT | 20.5 ± 2.8 (n = 9) | 20.2 ± 3.9 (n = 7) | 0.837 | 297.5 ± 94.3 (n = 11) | 264.7 ± 52.9 (n = 9) | 0.304 |
| 11DAT | 22.4 ± 5.1 (n = 15) | 20.4 ± 6.5 (n = 13) | 0.503 | 269.1 ± 79.6 (n = 15) | 294.2 ± 59.9 (n = 14) | 0.185 |
| 12DAT | 23.2 ± 5.8 (n = 13) | 21.5 ± 3.4 (n = 15) | 0.338 | 266.5 ± 96.4 (n = 13) | 262.3 ± 83.6 (n = 15) | 0.882 |

some experiments, we also quantified the number of transplant-derived cells during the period of cell death (*Figure 12B*). In data presented as the fraction of transplant-derived cells (*Figure 8* and *Figure 11*), GFP positive and tdTomato-positive cells were counted from coronal sections along the rostral-caudal axis in at least 10 sections per animal. The number of GFP or tdTomato-positive cells was divided by the total cell number (GFP + tdTomato) in that section. This fraction does not reflect the absolute number of cells, but their relative contribution to the overall population of transplant-derived cells at different DAT. For one experiment (*Figure 10*), we calculated the absolute number of transplant-derived WT (GFP) and *Pcdhg^{fcon3/fcon3}* (tdTomato) cells. For this experiment, 50 K cells of each genotype were counted before transplantation.

### Neuron Morphology analysis

Recipient brains were co-transplanted with *Gad1*-GFP (WT *Pcdhg*) and *Nkx2.1^{Cre}*;Ai14;*Pcdhg^{fcon3/fcon3}* (mutant *Pcdhg*) MGE cIN precursors. Transplanted cells were identified in sections (50 μm) stained for GFP and tdTomato and analyzed at 6, 8, 10 and 12 days after transplantation (DAT). Neuron morphology was reconstructed from confocal image (20X objective-4X zoom) stacks taken at 1 μm intervals, using Neurolucida software (MBF). Sections with a relatively low density of GFP+ (WT) and TdTomato+ (Mutant) transplant-derived neurons (in order to clearly image individual cells) were selected and (145.31 um²) fields tiled to cover all the visible processes. All GFP+ (WT) and TdTomato+ (Mutant) positive neurons in these tiles were reconstructed. Neuron morphometric analysis was done using Neurolucida Explorer.

**Table 2.** Ratios are reported for the frequency of spontaneous excitatory and inhibitory synaptic currents where wild type and *Pcdhg*<sup>fcon3/fcon3</sup> co-transplants are compared within the same slice.

The natural log of each ratio was averaged for 8, 9, 10, 11, and 12 DAT and show that slice variation was not significant.

| | Frequency of spontaneous excitatory synaptic currents | Frequency of spontaneous inhibitory synaptic currents |
|---|---|---|
| | LN($Pcdhg^{fcon3/fcon3}$/WT) | LN($Pcdhg^{fcon3/fcon3}$/WT) |
| | Mean ± SD | Mean ± SD |
| 8DAT | −0.305 ± 0.425 | 0.162 ± 0.793 |
| 9DAT | 0.177 ± 0.468 | 0.018 ± 0.550 |
| 10DAT | 0.321 ± 0.706 | 0.301 ± 0.436 |
| 11DAT | 0.120 ± 0.927 | −0.075 ± 0.262 |
| 12DAT | −0.523 ± 0.943 | −0.072 ± 0.128 |

### RT-PCR

Total RNA was prepared from dissected cortex of using Trizol (Invitrogen) and reverse-transcribed by Quantiscript Reverse Transcriptase (Qiagen), using a mix of oligo-dT and random primers, according to the manufacturer's protocol. Primer sequences for gene expression analysis in *Figure 1* are provided in *Appendix 1—table 1*.

### Slice electrophysiology

As in *Larimer et al., 2016*; *Larimer et al., 2017*; *Priya et al., 2019*, host animals were anesthetized and decapitated at 8, 9, 10, 11, or 12 days after transplant. The brain was removed into ice-cold dissection buffer containing (in mM): 234 sucrose, 2.5 KCl, 10 $MgSO_4$, 1.25 $NaH_2PO_4$, 24 $NaHCO_3$, 11 dextrose, 0.5 $CaCl_2$, bubbled with 95% $O_2$/5% $CO_2$ to a pH of 7.4. Coronal slices of visual cortex (200 µm thick) were cut via vibratome (Precisionary Instruments) and transferred to artificial cerebrospinal fluid (ACSF) containing (in mM): 124 NaCl, 3 KCl, 2 $MgSO_4$, 1.23 $NaH_2PO_4$, 26 $NaHCO_3$, 10 dextrose, 2 $CaCl_2$ (bubbled with 95% $O_2$/5% $CO_2$), incubated at 33°C for 30 min, then stored at room temperature. Fluorescently identified transplant-derived MGE-lineage interneurons (tdTomato +; *Pcdhg*<sup>fcon3/fcon3</sup> or GFP; WT) were viewed using IR-DIC video microscopy. Whole-cell current-clamp recordings were made with a Multiclamp 700B (Molecular Devices) using an internal solution that contained (in mM): 140 K-gluconate, 2 $MgCl_2$, 10 HEPES, 0.2 EGTA, 4 MgATP, 0.3 NaGTP, 10 phosphocreatine (pH 7.3, 290 mosm). Data were low-pass filtered at 2.6 kHz and digitized at 10 kHz by a 16 bit analog-to-digital converter (National Instruments). Data acquisition and analysis were done with custom software written in Matlab (https://www.mathworks.com/matlabcentral/fileexchange/21903-mphys). A Mann-Whitney (nonparametric) test, followed by multiple comparisons correction using the Benjamini-Hochberg stepdown method for control of false discovery rate (0.05 familywise) was used for the determination of statistical significance for all comparisons of physiological properties.

### Statistical Analysis

The person carrying quantifications was blinded to the genotype, except for data shown in *Figure 2—figure supplement 2*, and *Figure 3—figure supplements 3* and *4*. With the exception of slice electrophysiology data, all results were plotted and tested for statistical significance using Prism 8. All samples were tested for normality using the Shapiro-Wilk normality test. Unpaired comparisons were analyzed using the two-tailed unpaired Student's *t* test for normally distributed, and Mann-Whitney test for not normally distributed samples. For multiple comparisons analysis of one variable, either a one-way ANOVA with post hoc Tukey's test was used to compare the mean of each column with the mean of every other column, or a Dunnett test was used to compare the mean of each column to the mean of the control group for normally distributed samples. For samples with non-Gaussian distributions, a nonparametric Kruskal-Wallis test was performed followed by a post-hoc Dunn's test. Two-way ANOVA with post hoc Sidak's test was used for multiple comparisons with more than one variable. Outliers were identified using ROUT method with alpha set to 0.05.

## Acknowledgements

We thank Joshua Sanes for kindly proving the *Pcdhg^fcon3* mice. We also thank Ricardo Romero, Jose Rodrigues, and Cristina Guinto for technical help and John Rubenstein, Arnold Kriegstein, and Duan Xin for the insightful discussion of experimental data. This work was supported by NIH Grants (R01 NS028478, R01 EY02517) and a generous gift from the John G Bowes Research Fund to AAB; NIH Grant R01DC014101 to ARH, the Klingenstein Foundation to ARH, Hearing Research Inc to ARH, and the Coleman Memorial Fund to ARH; NIH Grants R01EY025174 and 5F32EY029935 to MPS and 5F32EY029935 to BR. AAB is the Heather and Melanie Muss Endowed Chair and Professor of Neurological Surgery at UCSF. MPS is a recipient of the Research to Prevent Blindness Disney Award for Amblyopia Research.

## Additional information

### Competing interests

Arturo Alvarez-Buylla: is cofounder, serves on the scientific advisory board, and owns shares in Neurona Therapeutics. The other authors declare that no competing interests exist.

### Funding

| Funder | Grant reference number | Author |
|---|---|---|
| National Institutes of Health | R01NS028478 | Arturo Alvarez-Buylla |
| National Institutes of Health | EY02517 | Arturo Alvarez-Buylla |
| National Institutes of Health | R01DC014101 | Andrea R Hasenstaub |
| National Institutes of Health | R01EY025174 | Michael P Stryker |
| National Institutes of Health | 5F32EY029935 | Benjamin Rakela Michael P Stryker |
| John G. Bowes Research Fund | | Arturo Alvarez-Buylla |
| The Klingenstein Foundation | | Andrea R Hasenstaub |
| Coleman Memorial Fund | | Andrea R Hasenstaub |
| Hearing Research institute | | Andrea R Hasenstaub |

The funders had no role in study design, data collection and interpretation, or the decision to submit the work for publication.

### Author contributions

Walter R Mancia Leon, Conceptualization, Formal analysis, Investigation, Writing - original draft, Writing - review and editing; Julien Spatazza, Conceptualization, Investigation; Benjamin Rakela, Formal analysis, Investigation, Visualization, Writing - review and editing; Ankita Chatterjee, Viraj Pande, Data collection; Tom Maniatis, Conceptualization, Resources; Andrea R Hasenstaub, Michael P Stryker, Conceptualization, Formal analysis, Supervision, Funding acquisition, Validation, Investigation, Visualization, Project administration, Writing - review and editing; Arturo Alvarez-Buylla, Conceptualization, Formal analysis, Supervision, Funding acquisition, Validation, Investigation, Visualization, Writing - original draft, Project administration, Writing - review and editing

### Author ORCIDs

Walter R Mancia Leon (iD) https://orcid.org/0000-0002-1920-6514
Andrea R Hasenstaub (iD) http://orcid.org/0000-0003-3998-5073
Arturo Alvarez-Buylla (iD) https://orcid.org/0000-0003-4426-8925

## Ethics

Animal experimentation: Data presented in this study were acquired following the University of California, San Francisco (UCSF) Institutional Animal Care Committee guidelines under the following protocols: AN178775-02C, AN180588, AN175872.

## Decision letter and Author response

Decision letter https://doi.org/10.7554/eLife.55374.sa1
Author response https://doi.org/10.7554/eLife.55374.sa2

# Additional files

## Supplementary files

• Transparent reporting form

## Data availability

Data generated for this study are included in the manuscript and source data files have been provided for Figures 1 to 13.

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

# Appendix

**Appendix 1—table 1. Clustered protocadherin primer sequences for RT-PCR gene expression analysis in _Figure 1_.**

| Gene name | Oligo name | Sequence | forward/reverse |
|---|---|---|---|
| Pcdhga1 | Pcdhga1 | CACGAGAGCTGTGAGAAACAGG | F |
| Pcdhga2 | Pcdhga2 | CTGATTTCCTCTCAGCACCTCAG | F |
| Pcdhga3 | Pcdhga3 | GAAACGAAAGAAGACCCCACGC | F |
| Pcdhga4 | Pcdhga4 | CTCCTGGTATCTCAAGACTTGC | F |
| Pcdhga5 | Pcdhga5 | CACACAAAGAAGAGCCCGGAGA | F |
| Pcdhga6 | Pcdhga6 | GCAAAGAGGAAGACTCTCTTGA | F |
| Pcdhga7 | Pcdhga7 | TCAAGAATGTAAGGGTGAAGCC | F |
| Pcdhga8 | Pcdhga8 | CATCCATAGATTTCCATGAGAATAA | F |
| Pcdhga9 | Pcdhga9 | TCAGTTGAGCCCAAGTTTCCT | F |
| Pcdhga10 | Pcdhga10 | CCAAGTGTCCTGTAGAAGACGC | F |
| Pcdhga11 | Pcdhga11 | GCGAGCCTCTCCTGATAACTG | F |
| Pcdhga12 | Pcdhga12 | CTTTTACCATCGGGTGATTCGG | F |
| Pcdhgb1 | Pcdhgb1 | CAGGATCTCCTGTGCGATGATC | F |
| Pcdhgb2 | Pcdhgb2 | GACTCTTGGGTACCAGGTACTC | F |
| Pcdhgb4 | Pcdhgb4 | TGATCAGTTGAAATCAGGACAAGA | F |
| Pcdhgb5 | Pcdhgb5 | CCTTCTTTGCCCTGAGTCATC | F |
| Pcdhgb6 | Pcdhgb6 | CTTAATTCCGCTTCACCTTGG | F |
| Pcdhgb7 | Pcdhgb7 | AAAGATAGCTCCTCGGCACTG | F |
| Pcdhgb8 | Pcdhgb8 | CGAGACCTTTGTACGGAAGC | F |
| Pcdhgc3 | Pcdhgc3 | GCTGCGAAGTTGTGATCCTGTG | F |
| Pcdhgc4 | Pcdhgc4 | CAAGCTGTCCACCCTCTGATCTT | F |
| Pcdhgc5 | Pcdhgc5 | GCCTTGCGTTCCCGCTCTAGTA | F |
| PcdhgCon | PcdhgCon | GTAAACTGGGGTCCGTATCGAG | R |
| Pcdhga1 | PcdhgA_1 | TTTTGTCAGCACCCCAGTC | F |
| Pcdhga2 | PcdhgA_2 | TTTCCTCTCAGCACCTCAGTC | F |
| Pcdhga3 | PcdhgA_3 | GTGGGAAAAGCGAGCCTCTTA | F |
| Pcdhga4 | PcdhgA_4 | AGCTGTGGGAAGAGTGATCC | F |
| Pcdhga5 | PcdhgA_5 | AGAGCTGTGAGAAGAGTGAGC | F |
| Pcdhga6 | Pcdh_γA6 | CATCAGTCAGGAGGGCTGTG | F |
| Pcdhga7 | PcdhgA_7 | ATCAGCCAAGATAGCTGTGAG | F |
| Pcdhga8 | 2 PcdhgA_8 | CATCCATAGATTTCCATGAG | F |
| Pcdhga9 | 2Pcdh_γA9 | TGTGGGAAGAGTGAACCTCTG | F |
| Pcdhga11 | 3Pcdh_γA11 | GAAAAGCGAGCCTCTCCTG | F |
| Pcdhgb1 | PcdhgB_1 | TGTGCGATGATCCTTCTGTG | F |
| Pcdhgb2 | 3Pcdh_γB2 | GACTCCGGAAGTTGCTCCTC | F |
| Pcdhgb4 | 2Pcdh_γB4 | CAGGACAAGATCTACAATTTGC | F |
| Pcdhgb5 | 4- PcdhgB5 | TCTGGACAAGGCCTTCTTTG | F |
| Pcdhgb6 | Pcdh_γB6 | GATCGTTTCCGGTAGTTCTCC | F |

_Appendix 1—table 1 continued on next page_

*Appendix 1—table 1 continued*

| Gene name | Oligo name | Sequence | forward/reverse |
|---|---|---|---|
| Pcdhgb7 | 2- PcdhgB7 | TCCAGCCGCACAAGATATTC | F |
| Pcdhgb8 | 2- PcdhgB8 | CGAGACCTTTGTACGGAAGC | F |
| Pcdhgc3 | PcdhgC_3 | TGCGAAGTTGTGATCCTGTG | F |
| Pcdhgc4 | PcdhgC_4 | CAAGCTGTCCACCCTCTGATC | F |
| PcdhgCon | Pcdhg_COM-R | GAGAGAAACGCCAGTCAGTG | R |
| Pcdha1 | 2-PcdhAlpha_1 | AAAGAAGTGACCACGCAGAAG | F |
| Pcdha2 | PcdhA2-F | GGAATCAGCAGAAGAGAGACAA | F |
| Pcdha3 | PcdhAlpha_3 | ACACCATGCCCAGTTAATCAAG | F |
| Pcdha4 | PcdhAlpha_4 | TCTGATTCAAGGGACAGAGAGG | F |
| Pcdha5 | Pcdh_α5 | TTCAGTCCCAGCCTACCTCA | F |
| Pcdha6 | Pcdh_α6 | TGAGCATCAGGATTTGAACG | F |
| Pcdha7 | Pcdh_α7 | GGGTCCCAGCTCTACAGATAAC | F |
| Pcdha8 | Pcdh_α8 | TTCTTTGGACTCCTCCGAGA | F |
| Pcdha9 | PcdhAlpha_9 | CCGAAGTGGGAATGGAAAGT | F |
| Pcdha10 | PcdhAlpha_10 | CAGTGTTCCTCCTGGTTTGG | F |
| Pcdha11 | Pcdh_α11 | TCCCAACCTGGGTAGAGATG | F |
| Pcdha12 | PcdhAlpha_12 | TGCAGAGGACACATGTCAGAG | F |
| Pcdhac1 | PcdhAlpha_C1 | TGCCAGTATCCTGTGTTCAGA | F |
| Pcdhac2 | Pcdh_αc2 | AACTCACCGGCCAAAGTAGG | F |
| PcdhaCon | Pcdh_α-CR | TGCTCTTAGCGAGGCAGAGTAG | R |
| Pcdhb1 | pcdhb1_F | TAGTTGCCGAGGGTAACAGG | F |
| Pcdhb1 | Pcdh_ß1_R | GGGTCAGATTTGCCACAAAG | R |
| Pcdhb2 | 2Pcdh_ß2_F | AATGGTTTGCTCCATCCAAG | F |
| Pcdhb2 | 2Pcdh_ß2_R | GATCCAGGGCTGTGTTTGTC | R |
| Pcdhb3 | PcdhB3-F | TCAGGGAAATGCACAGTCATAG | F |
| Pcdhb3 | PcdhB3-R | CACAGTTTCTGCTGAATTCTCG | R |
| Pcdhb4 | pcdhb4_F | TCTGGGATGACCACAGTTCA | F |
| Pcdhb4 | pcdhb4_R | TGCACCTCATAGAGCGATTG | R |
| Pcdhb5 | 3Pcdh_ß5_F | TGTGCTCACGCTCTACCTTG | F |
| Pcdhb5 | 3Pcdh_ß5_R | CTCACTCCCACGAACATCAG | R |
| Pcdhb6 | PcdhB6-F | GTCAAGGACAATGGAGAACCTC | F |
| Pcdhb6 | PcdhB6-R | CTTCATCCTGTGAAGAGTCGTG | R |
| Pcdhb7 | pcdhb7_F | GACCTCATGGAGAAGCTGGA | F |
| Pcdhb7 | 2-pcdhb7_R | GAGTTGTTGGCTCACTGCAA | R |
| Pcdhb8 | 2-pcdhb8_F | ATTCCATGCCAGAAGAAACG | F |
| Pcdhb8 | 2-pcdhb8_R | TAGTGGTCAGTTCCCCAACC | R |
| Pcdhb9 | 2-pcdhb9_F | AACAAAGGAAGCAGACAAGAGC | F |
| Pcdhb9 | 2-pcdhb9_R | TCACCGTGTTGCTCATAATCTC | R |
| Pcdhb10 | 2PcdhB10-F | GGTATTTGAGCGTGATCTAGGG | F |
| Pcdhb10 | 2PcdhB10-R | AGAGGGCGCTTCTTCTTCTAGT | R |
| Pcdhb11 | 3Pcdh_beta11_F | CGACCACTCTCCAGAGTTCC | F |
| Pcdhb11 | 3Pcdh_beta11_R | GCTGCCTTCAGAGGAAACAC | R |

*Appendix 1—table 1 continued on next page*

*Appendix 1—table 1 continued*

| Gene name | Oligo name | Sequence | forward/reverse |
|---|---|---|---|
| *Pcdhb12* | pcdhb12_F | CTGGGATATATGGCAATGTCG | F |
| *Pcdhb12* | pcdhb12_R | GTCAGACGGATTTCTCCTGTG | R |
| *Pcdhb13* | 2PcdhB13-F | GATAACGCTCCAGAAGTGATCC | F |
| *Pcdhb13* | 2PcdhB13-R | CTGCTGGTTATTTCCAGAGTCC | R |
| *Pcdhb14* | 2-pcdhb14_F | CCCAGCACACCATAACAGTG | F |
| *Pcdhb14* | 2-pcdhb14_R | GATGGTGCCTATGTGCAATG | R |
| *Pcdhb15* | PcdhB15-F | CTCAGTCCGCTTACTGGAGAAT | F |
| *Pcdhb15* | PcdhB15-R | AATGGTCTTTCCAACAGCAACT | R |
| *Pcdhb16* | pcdhb16-F | TCATCCGTGAGAACAACAGC | F |
| *Pcdhb16* | pcdhb16-R | GCAGCAGCGAGTAAGTGATG | R |
| *Pcdhb17* | PcdhB17-F | AAGAGAGCACTTGACAGGGAAG | F |
| *Pcdhb17* | PcdhB17-R | AGACCTGCACTGTTATGGTGTG | R |
| *Pcdhb18* | PcdhB18-F | TGCATGGAGTCATACTTTGGAC | F |
| *Pcdhb18* | pcdhb18-R | TAGCCATGTTTAGAAAGCAGCA | R |
| *Pcdhb19* | 3Pcdh_beta19_F | TTCGCTCTTCCTCCTCTCTG | F |
| *Pcdhb19* | 3Pcdh_beta19_R | AGTTCCCGCACTGTTAATGC | R |
| *Pcdhb20* | PcdhB20-F | GAAGTGATCATGTCGTCGGTTA | F |
| *Pcdhb20* | PcdhB20-R | CTTCCGTTGTCTCCAGAGTCTT | R |
| *Pcdhb21* | 2Pcdh_beta21_F | ACAGCACTCGGGCTTTCTAC | F |
| *Pcdhb21* | 2Pcdh_beta21_R | GGCAGCTCAGAGAGTGGTTC | R |
| *Pcdhb22* | 2Pcdh_beta22_F | GCTCTGCTAGCGTCACACTG | F |
| *Pcdhb22* | 2Pcdh_beta22_R | ATCACCTTCCTGGTGACTGG | R |

