## [Decision Letter]

**Acceptance summary:**

As acknowledged by the three reviewers, your study highlights a novel and exciting role for this family of protocadherins in the death of cortical interneurons, by combining global genetic models and in vivo transplants. The revised version of the manuscript addresses most of the comments raised by reviewers. It provides a novel perspective on how the density of cortical interneurons is regulated together with activity-dependent processes.

**Decision letter after peer review:**

Thank you for submitting your article "Clustered γ-Protocadherins Regulate Cortical Interneuron Programmed Cell Death" for consideration by *eLife*. Your article has been reviewed by three peer reviewers, and the evaluation has been overseen by a Reviewing Editor and Marianne Bronner as the Senior Editor. The following individual involved in review of your submission has agreed to reveal their identity: Gordon Fishell (Reviewer #1); Debra L Silver (Reviewer #2).

The reviewers have discussed the reviews with one another and the Reviewing Editor has drafted this decision to help you prepare a revised submission. In recognition of the fact that revisions may take longer than the two months we typically allow, until the research enterprise restarts in full, we will give authors as much time as they need to submit revised manuscripts.

This elegant study provides novel insights onto the mechanisms regulating the death of interneurons, which contributes to the proper functioning of cortical circuits and potentially to the etiology of neurodevelopmental disorders. The use of multiple experimental approaches reveals the roles of *Pcdh-*γ in regulating the survival of interneurons during the endogenous period of programmed cell death.

While all reviewers highlighted the quality and interest of the work, they also raised a numbers of concerns that you should be able to address without additional experimental manipulations.

Reviewer #1:

This is a very nicely done paper by the group that first identified the rather large extent of cell death that occurs in interneuron populations derived from the MGE. I on the whole think it is a very solid paper well suited to publication in Life. I however have a number of queries I would like to see addressed.

The major thing I am confused about is the autonomous/non-autonomous actions of Cy3,4,5 protocadherins. While the authors do a nice job of ruling out an action of the α and β subforms and reverse the affect on cell death by crossing into the BAX background, I can't see how they imagine how the non-autonomous effects of mutant cells could have when transplanted with wild type cells they are co-injected with. In short they claim 92% autonomous effect and a 62% non-autonomous effect but how is that possible. Surely right after transplant the wild type cells interact with the host wild type cells not the mutant ones and given their argument that the migration and settling are unaffected by the loss of these isoforms, it is hard to imagine why they see such a pronounced non-autonomous effect. At very least they need to provide a plausible explanation.

They do a nice job up front showing that the CY3-5 isoforms are the ones most expressed during cell death but it is clear that the CY5 form is the one highest expressed. It would be nice to look at whether the expression of CY5 or all forms is affected in wild type cells when transplanted with the mutant cells with qPCR

Multiple papers (Burrone and Marin as referenced and Priya et al., which is not) all point to activity playing a role in cell death in interneurons. The authors look at the intrinsic properties in the mutant cells that survive and find no alterations, but the data clearly supports a like between activity and cell death. Ideally they would explore whether altering activity in interneurons with DREADDS or Kir2.1 affects protocadherin expression but at very least they should comment on the possible connection.

Reviewer #2:

General Assessment of the Work: Overall, this paper presents evidence for *Pcdh-*γ as required for cIN survival during the endogenous period of PCD. The authors use a number of genetic mouse models to show that knocking out *Pcdh-*γ conditionally in cINs leads to a 50% reduction in surviving cINs at p30 without influencing proliferation and migration at embryonic stages. Additionally, they show that these findings are specific to *Pcdh-*γ and do not overlap with Pcdh-a or Pcdh-B. While it does promote cell death, knockdown of *Pcdh-*γ does not impair cIN morphology or physiological properties. Overall this work is interesting, technically elegant, and well performed. I have a few suggestions to better support their findings.

Major Concerns:

1) In Figure 5C the authors use Bax^-/-^; Pcdh^-/-^ mice to show these contain far more cINs than their WT counterparts. However, this experiment is missing the Bax^-/-^ alone control. It is insufficient to just refer to a prior paper as these experiments may have been performed by a different investigator, who has a different quantification baseline, etc.

2) The finding that survival of WT cINs is reduced in the presence of *Pcdh-*γ mutant cells is interesting. However, the evidence for this non-cell autonomous effect could be better strengthened. First they should include statistical comparisons comparing the reported fold changes in the presence and absence of the mutant cells (ie. Comparing data in Figures E and C). This observation would be strengthened with an experiment examining the survival of WT cINs transplanted into WT brains vs *Pcdh-*γ cKO brains.

3) I found the results of some co-transplantation experiments with WT cells confusing. In Figure 8B co-transplantation of two fluorescent WT cells, the authors show that there is no reduction of cINs from 3DAT to 21DAT. However, based upon this work and that of Southwell et al., I would have expected some death by this stage of WT alone (since the intrinsic age of these cells would coincide with peak of programmed cell death). In Figure 8B' with co-transplantation of WT and mutant cells, why is the survival fraction of WT cells now increasing from 6DAT to 13DAT and 21DAT? Similarly, in Figure 11B co-transplantation experiments, the survival fraction of WT cells is constant, but in Figure 11C it is increasing. These contradictory findings need to be addressed as they introduce concern about the validity of some of these transplantation experiments. The authors should also consider representing data not as survival fraction (which is not even defined in the Materials and methods) but instead as number of cells in the cortex (as they do in Figure 10C, E).

Reviewer #3:

In "Clustered γ-Protocadherins regulate cortical interneuron programmed cell death", Mancia et al. provide clear and solid evidence about the function of γ-Protocadherins and more specifically that of γC3- γC4- γC5 isoforms in cortical interneuron (cIN) programmed cell death (PCD). The findings are in agreement with what it has been previously reported in other systems (spinal cord, retina) and although the authors provide no mechanism about the role of Pcdh-γ cluster in cIN PCD, the results are very interesting and they substantially enhance our knowledge on how the final number of cIN might be adjusted.

There are a couple of issues the authors should further elaborate/explain:

1) One of the most interesting results of the paper came up from their cell transplantation experiments. The authors have used this technique (as it has been done in the past from the same group; Southwell et al., 2012) to investigate the survival of WT and different types of Pcdh- mutant cells in the same conditions (environment). In one of these experiments they show that by grafting MGE INs that comprise 50% Pcdh-γ mutant cells, the survival of the WT cells is decreased by approximately 20% compare to grafts consisted of 100% WT cells. This is suggestive of a non-cell autonomous mechanism of Pcdh-γ on cIN PCD. Taking into consideration that the Nkx2.1CRE line, used in this paper, labels only a percentage of the MGE cINs according to the original publication (Figure 6 in Xu et al., 2008), and others as well, it would be interesting to see what is happening within this mixed MGE population in control and mutant Nkx2.1CRE;Pcdh-γ mice. Do PV and SST escapers (ai14-negative; CRE-negative) survive less? This is particularly interesting since a recent paper in BioRxiv (Carriere et al., 2020) shows the opposite results (increased number of WT MGE cINs in the Pcdh-γ mutants). If this is the case and this result is verified, how do the authors explain the difference between heterochronic cell-transplantations and the homochronic cIN survival analysis?

2) Concerning the same experiment, in the Discussion, they speculate that the increase in the WT cIN cell-death in the above transplantations, is related to the fact that WT cells fail to interact with other cINs of the same age and initiate an intracellular Pcdh-γ pro-survival signalling cascade. Therefore, is it a matter of percentages? What is expected if they try transplantations consisting of 10%, 25% or 75% mutant cells? And anyway, how do we know that is due to a failure of cell-cell interaction and not due to a neurotoxic effect of a big number of neurons dying, in which young INs of the same age are more vulnerable? We believe that the above experiments will further clarify their results and they are essential since there is no proposed mechanism.

3) The Discussion is quite confusing. The authors suggest at least 3 potential mechanisms: 1) Population-cell autonomous mechanism. This model is based on cell-cell interactions between cINs, which is important for having the number of INs "computed" before arriving in the cortex (?). This is not explained very well. 2) A role, specifically for Pcdh-γC4, as it has been recently suggested that this is the isoform that mediates cell death (Garrett et al., 2019). Interestingly, this isoform might not be found on the cell-membrane, therefore it might not be implicated in cell-cell interactions (Garrett et al., 2019). 3) Finally, pyramidal neurons and interneurons may interact via the Pcdh-γ cluster (two possibilities presented), therefore a non-population autonomous mechanism. The authors should present all these possibilities but in a clear more constructive manner. In addition, they do not discuss at all, any potential pro-survival mechanisms that have been shown and might apply in cIN PCD as well. For example, in a recent paper (Duan et al., 2020) GABA signalling is suggested to be important for the survival of MGE derived INs (mentioned in the Discussion). Pcdh-γ-C5 isoform it has been shown to interact with GABAA receptors, stabilizing GABAergic synapses (Li et al., 2012). Can the authors comment on this?

---

## [Author Response]

Reviewer #1:This is a very nicely done paper by the group that first identified the rather large extent of cell death that occurs in interneuron populations derived from the MGE. I on the whole think it is a very solid paper well suited to publication in Life. I however have a number of queries I would like to see addressed.The major thing I am confused about is the autonomous/non-autonomous actions of Cy3,4,5 protocadherins. While the authors do a nice job of ruling out an action of the α and β subforms and reverse the affect on cell death by crossing into the BAX background, I can't see how they imagine how the non-autonomous effects of mutant cells could have when transplanted with wild type cells they are co-injected with. In short they claim 92% autonomous effect and a 62% non-autonomous effect but how is that possible. Surely right after transplant the wild type cells interact with the host wild type cells not the mutant ones and given their argument that the migration and settling are unaffected by the loss of these isoforms, it is hard to imagine why they see such a pronounced non-autonomous effect. At very least they need to provide a plausible explanation.

Since the concept of cell-autonomy was confusing and we present evidence for both cell autonomous and non-cell autonomous mechanisms, we have eliminated this wording from the text and instead now describe directly our observations:

In the Results section:

“Given these observations, we next asked how the presence of γ-Pcdh mutant cIN affected the survival of WT cIN in the transplantation setting. We co-transplanted 50K Gad67-GFP γ-Pcdh WT (GFP+) with 50K Nkx2.1::Cre;Ai14 γ-Pcdh mutant (tdTomato+) MGE cIN precursors and compared the survival of each population at 6 and 21 DAT. At 6 DAT the total number of tdTomato+ cells in the cortex of recipient mice was similar to that of GFP+ cells (Figure 10A, D and E). However, between 6 and 21DAT, the total number of GFP+ cells had decreased by an average of ~63% (Figure 10E, compared to Figure 10C). Compared to the ~40% of endogenous or transplanted WT cINs that are normally eliminated (present study and previous work (Southwell et al., 2012; Wong et al., 2018)), this experiment suggests that WT cells die at a higher rate (63%) when co-transplanted with γ-Pcdh mutant MGE cells. However, this observation would require additional animals for statistical confirmation. Regardless, the number of γ-Pcdh mutant (tdTomato+ cells) cINs decreased dramatically, by ~96% (Figure 10E). This experiment confirms that MGE cells lacking γ-Pcdh function are eliminated in far greater numbers than control MGE cells and show that the presence of γ-Pcdh WT cINs within a mixed population also affects the survival of mutant cINs (compare Figure 8 and Figure 10). These observations are consistent with the hypothesis that cINs interact with others of the same age discussed below.*”*

Our interpretation in the Discussion:

“Interestingly, the survival of WT cIN may also be reduced when co-transplanted with γ-Pcdh deficient MGE-cells although this difference did not reach statistical significance with the numbers of cases studied. If true, these findings would be consistent with the notion that cell-cell interactions among young cIN after their migration, is an essential step in determining their final numbers. However, we cannot exclude that the increase in the elimination of WT cells may result from a non-specific (e.g., toxic) effect of the increased cell death among γ-Pcdh mutant cells. If the latter occurs, the process is specific to the population of Nkx2.1^+^ MGE-derived cINs because there was no effect on cell death of WT CGE-derived VIP cINs (Figure 3E, Figure 3—figure supplement 1) or on non-Nkx2.1-derived SST or PV cells (Figure 3—figure supplement 4). Interactions mediated by γ-Pcdhs, and specifically among the C-isoforms, may directly or indirectly regulate survival of cINs of the same age and origin”

They do a nice job up front showing that the CY3-5 isoforms are the ones most expressed during cell death but it is clear that the CY5 form is the one highest expressed. It would be nice to look at whether the expression of CY5 or all forms is affected in wild type cells when transplanted with the mutant cells with qPCR

This is an interesting experiment that could begin addressing the mechanism by which γ-Pcdhs regulate cIN survival. However, this is not a simple experiment; γ-Pcdhs, including isoforms γC3, γC4, and γC5 are expressed at low levels and their detection specifically in the subpopulation of WT or Mutant cells in our co-transplants, would require developing new methods to cleanly isolate WT cells. This will take at least a year and seems beyond the scope of the current MS: identifying the role that specific PCDHs play in cIN survival.

Multiple papers (Burrone and Marin as referenced and Priya et al., which is not) all point to activity playing a role in cell death in interneurons.

We apologize for this oversight; we now include the reference to Priya et al.

The authors look at the intrinsic properties in the mutant cells that survive and find no alterations, but the data clearly supports a like between activity and cell death. Ideally they would explore whether altering activity in interneurons with DREADDS or Kir2.1 affects protocadherin expression but at very least they should comment on the possible connection.

We do not know of any evidence of clustered Pcdhs being regulated by neuronal activity. This is an interesting suggestion for a follow-up study. Since clustered Pcdhs are expressed at very low levels this would require stringent controls to determine the expression per cell. This seems to be well beyond the scope of our study which focuses on the identification of γ-Pcdhs as key in the regulation of cell death.

Reviewer #2:General Assessment of the Work: Overall, this paper presents evidence for Pcdh-y as required for cIN survival during the endogenous period of PCD. The authors use a number of genetic mouse models to show that knocking out Pcdh-y conditionally in cINs leads to a 50% reduction in surviving cINs at p30 without influencing proliferation and migration at embryonic stages. Additionally, they show that these findings are specific to Pcdh-y and do not overlap with Pcdh-a or Pcdh-B. While it does promote cell death, knockdown of Pcdh-y does not impair cIN morphology or physiological properties. Overall this work is interesting, technically elegant, and well performed. I have a few suggestions to better support their findings.Major Concerns:1) In Figure 5C the authors use Bax^-/-^; Pcdh^-/-^ mice to show these contain far more cINs than their WT counterparts. However, this experiment is missing the Bax^-/-^ alone control. It is insufficient to just refer to a prior paper as these experiments may have been performed by a different investigator, who has a different quantification baseline, etc.

We had not included this control as the rescue previously reported for Bax^-/-^ mice (Southwell et al., 2012)) was very similar to that observed in the Bax^-/-^; γ-Pcdh^fcon3/fcon3^ mice. However, to have this analysis done by the same person, we now include control animals that were Bax^-/-^ but γ-Pcdh^fcon3/+^ (heterozygous for γ-Pcdh). This data has been added to Figure 5C. The number of cIN in Bax^-/-^ ;γ-Pcdh^fcon3/+^ mice was very similar to that observed in Bax^-/-^ ;γ-Pcdh^fcon3/fcon3^ mice.

In the Results section:

“Importantly, the homozygous deletion of the pro-apoptotic BCl^-^2-associated X protein (BAX) rescued cIN density in the γ-Pcdh mutant mice to levels similar to those observed in control BAX^-/-^ ;Pcdh^fcon3/+^ mice or in mice carrying only the BAX mutation (BAX^-/-^) (Southwell et al., 2012) (Figure 5C )”

2) The finding that survival of WT cINs is reduced in the presence of Pcdh-y mutant cells is interesting. However, the evidence for this non-cell autonomous effect could be better strengthened. First they should include statistical comparisons comparing the reported fold changes in the presence and absence of the mutant cells (ie. Comparing data in Figures E and C). This observation would be strengthened with an experiment examining the survival of WT cINs transplanted into WT brains vs Pcdh-y cKO brains.

We have compared the fold change in the number of cINs between 6 and 21 DAT when transplanting WT-WT or WT-γ-Pcdh^fcon3/fcon3^ cINs. In the case of WT-WT transplants we observed a 39% drop in cIN number. For the WT-γ-Pcdh^fcon3/fcon3^ transplant we observed that WT dropped 63% and the γ-Pcdh^fcon3/fcon3^ cells dropped by 96%. This suggests that cell death among WT cINsis increased when mutant cells are present. Clearly 63% death is very different to the elimination of ~40% for endogenous WT cells (Southwell et al., 2012) or for transplanted WT cells in the present study. However, statistical analysis would require larger n (Mann Whitney test with current data comparing the relative change from 6 to 21 DAT in the WT population from WT-WT co-transplants to the relative change from 6 to 21 DAT in the WT population from WT-γ-Pcdh^fcon3/fcon3^ co-transplants shows a p value of 0.0519. We explain in the MS. that this is a suggestion and further experiments would be required to clearly determine how WT cIN cell death is affected by the presence of the γ-Pcdh^fcon3/fcon3^ cINs. The suggested experiment of transplanting WT cells into γ-Pcdh mutant environment is an interesting one, but not easy to interpret. We assume the reviewer was referring to transplantation into the Nkx2.1::Cre;γ-Pcdh^fcon3/fcon3^ mice as the straight γ-Pcdh^fcon3/fcon3^ mice die at birth. However, in this heterochronic transplantation, unlike our co-transplantation experiment, the transplanted E13.5 WT cINs would be a few days younger than the host mutant cINs (P2-6) and therefore at a different stage of their maturation. It would be interesting to transplant homochronically E13.5 WT cells into E13.5 Nkx2.1::Cre;γ-Pcdh^fcon3/fcon3^ embryos, but this experiment would require several months as it requires transplantations into the embryos.

3) I found the results of some co-transplantation experiments with WT cells confusing. In Figure 8B co-transplantation of two fluorescent WT cells, the authors show that there is no reduction of cINs from 3DAT to 21DAT. However, based upon this work and that of Southwell et al., I would have expected some death by this stage of WT alone (since the intrinsic age of these cells would coincide with peak of programmed cell death). In Figure 8B' with co-transplantation of WT and mutant cells, why is the survival fraction of WT cells now increasing from 6DAT to 13DAT and 21DAT? Similarly, in Figure 11B co-transplantation experiments, the survival fraction of WT cells is constant, but in Figure 11C it is increasing.

We apologize for this confusion; Indeed the number of both WT and γ-Pcdh^fcon3/fcon3^ mutant cells decreases in these co-transplantations. Our plots represent the fraction, from all transplant-derived cells (WT+ γ-Pcdh^fcon3/fcon3^ mutant cINs), that are WT or mutant at the different times after transplantation. Although the total number of both WT and γ-Pcdh^fcon3/fcon3^ mutant cells decreases, with increasing survival after transplantation, the fraction of cells that are WT becomes progressively higher compared to the fraction of mutant cells. In the co-transplantation of WT-WT, both populations decrease at equal rates and therefore at the different times they maintain similar proportions. This has now been clarified in the text (Results and Materials and methods sections) and in the first figure legend where this type of plots are presented. We have also added a supplementary figure to Figure 8 (Figure 8—figure supplement 1) where the absolute numbers of WT and γ-Pcdh^fcon3/fcon3^ mutant cells are plotted. The calculation to determine survival fraction is defined in Materials and methods. In supplementary Figure 1 to Figure 10 we also present the absolute numbers for cIN cell death in the co-transplantation of: WT-WT and WT-mutant cells.

In the Results section:

“While equivalent numbers of red and green cells were mixed before being transplanted, the absolute number of cells transplanted varied from transplant to transplant. In order to compare the survival, we use the fraction of green or red cells, among all co-transplanted cells (red+green). The fraction of surviving GFP+ and tdTomato+ cells at 3, 6, 13 and 21 days after transplantation (DAT) was measured (Figure 8A and B, top graph). ”

“ However, the fraction of γ-Pcdh mutant cINs (tdTomato+) surviving was dramatically lower when the transplanted cells reached a cellular age equivalent to that of endogenous cIN after the normal wave of programmed cell death (6DAT is roughly equivalent to P0; 21DAT is roughly equivalent to P15) (Southwell et al., 2012). Note that in this experiment the proportion of WT cells increases during this same period. This change in proportion is not a reflection of increased survival, as these cells also undergo elimination by programmed cell death (see below)(Figure 8—figure supplement 1), but that, with the increased loss of mutant cells, the WT cells account for a larger fraction of the total.”

In the material and methods section:

“In some experiments the absolute number of grafted cells varies between transplants; hence we report findings as the fraction of cells (GFP+ or tdTomato+) that survive from the total transplant-derived cell number in that animal (GFP + tdTomato-positive) (Figure 8). We determined for these experiments the number of transplant-derived cells of the different genotypes before and, in different animals, after the period of cell death. For some experiments we also quantified the number of transplant-derived cells during the period of cell death (Figure 12B). In data presented as the fraction of transplant-derived cells (Figure 8 and Figure 11), GFP positive and tdTomato-positive cells were counted from coronal sections along the rostral-caudal axis in at least 10 sections per animal. The number of GFP or tdTomato-positive cells was divided by the total cell number (GFP + tdTomato) in that section. This fraction does not reflect the absolute number of cells, but their relative contribution to the overall population of transplant-derived cells at different DAT. For one experiment (Figure 10), we calculated the absolute number of transplant-derived WT (GFP) and γ-Pcdh^fcon3/fcon3^ (tdTomato) cells. For this experiment, 50K cells of each genotype were counted before transplantation.”

These contradictory findings need to be addressed as they introduce concern about the validity of some of these transplantation experiments. The authors should also consider representing data not as survival fraction (which is not even defined in the Materials and methods) but instead as number of cells in the cortex (as they do in Figure 10C, E).

This has been clarified above. Note that in Figure 10, we present absolute numbers of transplant derived cINs. For this more stringent experiment we counted the number of cells transplanted and made sure that equal numbers of mutant and WT cells were transplanted. This has now been clarified as indicated above in the Materials and methods and Results’ sections.

Reviewer #3:In "Clustered γ-Protocadherins regulate cortical interneuron programmed cell death", Mancia et al. provide clear and solid evidence about the function of γ-Protocadherins and more specifically that of γC3- γC4- γC5 isoforms in cortical interneuron (cIN) programmed cell death (PCD). The findings are in agreement with what it has been previously reported in other systems (spinal cord, retina) and although the authors provide no mechanism about the role of Pcdh-γ cluster in cIN PCD, the results are very interesting and they substantially enhance our knowledge on how the final number of cIN might be adjusted.There are a couple of issues the authors should further elaborate/explain:1) One of the most interesting results of the paper came up from their cell transplantation experiments. The authors have used this technique (as it has been done in the past from the same group; Southwell et al., 2012) to investigate the survival of WT and different types of Pcdh- mutant cells in the same conditions (environment). In one of these experiments they show that by grafting MGE INs that comprise 50% Pcdh-γ mutant cells, the survival of the WT cells is decreased by approximately 20% compare to grafts consisted of 100% WT cells. This is suggestive of a non-cell autonomous mechanism of Pcdh-γ on cIN PCD. Taking into consideration that the Nkx2.1CRE line, used in this paper, labels only a percentage of the MGE cINs according to the original publication (Figure 6 in Xu et al., 2008), and others as well, it would be interesting to see what is happening within this mixed MGE population in control and mutant Nkx2.1CRE;Pcdh-γ mice. Do PV and SST escapers (ai14-negative; CRE-negative) survive less? This is particularly interesting since a recent paper in BioRxiv (Carriere et al., 2020) shows the opposite results (increased number of WT MGE cINs in the Pcdh-γ mutants). If this is the case and this result is verified, how do the authors explain the difference between heterochronic cell-transplantations and the homochronic cIN survival analysis?

We have now performed the quantification of the un-recombined PV and SST cells and consistent with (Carriere et al., 2020) we see a significant increase in the density of PV+/Ai14- and SST+/Ai14- cells in Nkx2.1::Cre;Ai14;γ-Pcdh^fcon3/fcon3^ mice. Most likely these un-recombined cells are derived from the most dorsal MGE at the interface with LGE (Hu, et al., 2017; Fogarty et al., 2007; Sousa et al., 2009). This region does not express, or expresses very low levels, of NKX2.1, but expresses NKX6.2 (Sousa et al., 2009; Fogarty et al., 2007) and generates a subpopulation of SST and PV cINs. We do not know if the presence of the conditional γ-Pcdh^fcon3^ allele results in increased production of these cells or if un-recombined cells from this domain increase their survival to compensate for the loss of Nkx2.1-cIN lacking γ-Pcdh function. If the latter is true it is indeed interesting that the behavior of these cells differs from that observed for WT cells co-transplanted with MGE cells lacking γ-PCDH function. Understanding these differences will require additional experiments to determine if cell-birth, migration or survival is affected among un-recombined PV and SST cINs. Since in our transplants we can only see the recombined cells, an additional reporter for all MGE cells (Nkx2.1 and Nkx6.2) will be required to study the phenomenon using transplantation. We have added a description of these findings in the Results section and a brief comment in the Discussion.

In Results:

“Interestingly, we observed that in our *Nkx2.1::Cre;Ai14; γ-Pcdh^fcon3/fcon3^* mice, the number of un-recombined PV and SST (PV+/Ai14- and SST+/Ai14-) cells was significantly increased compared to WT mice (Figure 3—figure supplement 4), a result consistent with recent observations (Carriere et al., 2020). PV+/Ai14- and SST+/Ai14- cells are likely derived from the most dorsal MGE at the interface with LGE expressing Nkx6.2 in a region of low, or no expression of Nkx2.1 (Hu et al., 2017; Fogarty et al., 2007; Sousa et al., 2009; Hu et al., 2017). We do not know if the presence of the conditional FCON3 allele results in increased production of these cells or if un-recombined cells from this domain increase their survival in compensation for the loss of cIN that lack Pcdh-γ function. If the latter is true, the behavior of these un-recombined PV and SST cINs differs from that observed for WT cells co-transplanted with MGE cells lacking Pcdh-γ function (see below).”

In the Discussion:

“Consistent with recent observations (Carriere et al., 2020), the number of un-recombined PV and SST (PV+/tdTomato- and SST+/tdTomato-) cINs in *Nkx2.1::Cre;Ai14; γ-Pcdh^fcon3/fcon3^* mice, increased compared to WT mice (Figure 3—figure supplement 4). These cells, which are likely derived from the dorsal Nkx6.2+/Nkx2.1- MGE domain (Hu et al., 2017; Fogarty et al., 2007; Sousa et al., 2009; Hu et al., 2017) may increase their survival in compensation for the loss of Nkx2.1-derived cINs lacking Pcdh-γ function. However, we cannot exclude that the increased number of un-recombined PV and SST cells in *Nkx2.1::Cre;Ai14; γ-Pcdh^fcon3/fcon3^* mice resulted from increased production or migration of cIN derived from regions of low, or no, expression of Nkx2.1. Further experiments will be required to understand the origin of these un-recombined PV+/Ai14- and SST+/Ai14- cINs and whether the observed increase in their numbers is due to compensatory survival mechanisms.”

2) Concerning the same experiment, in the Discussion, they speculate that the increase in the WT cIN cell-death in the above transplantations, is related to the fact that WT cells fail to interact with other cINs of the same age and initiate an intracellular Pcdh-γ pro-survival signalling cascade. Therefore, is it a matter of percentages? What is expected if they try transplantations consisting of 10%, 25% or 75% mutant cells? And anyway, how do we know that is due to a failure of cell-cell interaction and not due to a neurotoxic effect of a big number of neurons dying, in which young INs of the same age are more vulnerable? We believe that the above experiments will further clarify their results and they are essential since there is no proposed mechanism.

This is correct; we cannot distinguish between decreased interactions or increased toxicity as a by-product of increased dying cells among the mutant cells. We now make it clear that the decreased interaction, as a result of the presence of mutant cells, is a hypothesis and that other interpretations (e.g. toxic effect from the dying mutant cells) are possible. We have performed experiments transplanting different proportions of mutant and WT cells (see graphs in Author response image 1), as suggested by this reviewer, and see that the number of dying WT cells in a co-trasnplant in which the majority (80%) of cells are mutant (80 Mutant: 20 WT) appears (by the slope of the change between 3 and 21 DAT) to be similar to that when transplant is 50:50. However, this experiment is preliminary and many more transplants would be required to make an accurate comparison. While the experiment is interesting and would suggest that the presence of mutant cells in any number equally conditions the survival of WT cells, this experiment does not allow us to distinguish between the toxic or interaction hypotheses.

3) The Discussion is quite confusing. The authors suggest at least 3 potential mechanisms: 1) Population-cell autonomous mechanism. This model is based on cell-cell interactions between cINs, which is important for having the number of INs "computed" before arriving in the cortex (?). This is not explained very well. 2) A role, specifically for Pcdh-γC4, as it has been recently suggested that this is the isoform that mediates cell death (Garrett et al., 2019). Interestingly, this isoform might not be found on the cell-membrane, therefore it might not be implicated in cell-cell interactions (Garrett et al., 2019). 3)

The Discussion has been clarified and the speculation that cIN numbers may be “computed” before cINs arrive in the cortex has been eliminated. It is indeed interesting that γ-Pcdh C4 may be key to the regulation of interneuron survival, not only the spinal cord (Garrett et al., 2019) but also in other parts of the CNS (this is briefly discussed). These mice were made available to us through the laboratory of Joshua Weiner before the lock-dawn, but this transfer is now delayed. Our understanding is that γC4 is in the membrane, but unlike other PCDHs it requires binding to b PCDHs in the Α or Gama gene clusters to be transported to the membrane (Thu et al., 2014).

In the Discussion:

“A recent study suggests that the 𝛾C4 isoform is the key mediator in the regulation of neuronal cell death in the spinal cord (Garrett et al., 2019). How the specific 𝛾C4 isoform in the γ-Pcdh cluster mediates cell death remains a fundamental question for future research. Interestingly, the 𝛾C4 isoform appears to be unique in that it is the only Pcdh isoform that does not bind in a homophilic manner (Garrett et al., 2019) and it is not translocated to the membrane unless it is associated with other α- or γ-Pcdhs (Thu et al., 2014). The mechanism by which the 𝛾C4 isoform regulates cell-cell interactions among young cINs, leading to the adjustment of local circuit neuron numbers, remains unclear.”

Finally, pyramidal neurons and interneurons may interact via the Pcdh-γ cluster (two possibilities presented), therefore a non-population autonomous mechanism. The authors should present all these possibilities but in a clear more constructive manner. In addition, they do not discuss at all, any potential pro-survival mechanisms that have been shown and might apply in cIN PCD as well. For example, in a recent paper (Duan et al., 2020) GABA signalling is suggested to be important for the survival of MGE derived INs (mentioned in the Discussion). Pcdh-γ-C5 isoform it has been shown to interact with GABAA receptors, stabilizing GABAergic synapses (Li et al., 2012). Can the authors comment on this?

Thank you for this suggestion. We have re-organized the Discussion as follows.

In the Discussion:

“The co-transplantation assay, implemented in the present study, provides strong evidence that γ-Pcdhs in cIN are key to their selection by programmed cell death. γ-Pcdhs could be mediating initial cell-cell interactions that are important for the survival of cINs. Two non-exclusive possibilities exist: (1) γ-Pcdh mediate cell-cell interaction among young cIN to adjust their population size, and levels of inhibition, according to the numbers that reached the cortex; (2) γ-Pcdh mediate interactions with locally produced excitatory pyramidal neurons to adjust final numbers according to local levels of excitation. For the latter, MGE-derived cIN could interact with pyramidal neurons via γ-Pcdh C-isoforms. However, alternative # 2 is unlikely to explain how γ-Pcdhs adjust cIN numbers; data using conditional removal of γ-Pcdh in pyramidal cells show no effect on the survival of cINs (Carriere et al., 2020). However, we cannot exclude that initial connectivity with excitatory pyramidal neurons may indeed require the proper expression of γ-Pcdh among cIN through non-homophilic interactions.”

“A recent study has shown that coordinated activity of synaptically connected assemblies of cINs is essential for their survival (Duan et al., 2020). Pyramidal cells receive information from these assemblies via GABA_A_γ2-signaling and through the de-synchronization of their activity regulate cIN program cell death. γ-Pcdhs could be important in bringing together cINs of a common origin and at similar stages of maturations for the formation of initial cINs functional assemblies. The formation of these assemblies of synchronously firing cIN and the subsequent selection by pyramidal driven de-synchronization could explain both cell/population autonomous (Southwell et al., 2012)), and non-cell autonomous (Wong et al., 2018) mechanism of cIN programmed cell death. Interestingly, the Pcdh isoform γC5 binds to the GABA_A_γ subunit of the GABA receptor (Li et al., 2012), but the role of γC5-GABA_A_γ interaction on neuronal survival remains unknown. The transplantation assay provides a powerful tool to further study how γ-Pcdhs, cell-cell interactions, and cellular age contribute to cIN selection. It will be interesting, for example, to determine if heterochronically transplanted cINs form functional assemblies and whether these assemblies are affected by the removal of different γ-Pcdhs.”